

**SEALDH-II – a calibration-free transfer standard for airborne water vapor measurements:**
**Pressure dependent absolute validation from 5 – 1200 ppmv**
**at a metrological humidity generator**
**Bernhard Buchholz[1,2,4], Volker Ebert[1,2,3]**
[1] *Physikalisch-Technische Bundesanstalt Braunschweig, Germany*
[2] *Physikalisch Chemisches Institut, Universität Heidelberg, Germany*
[3] *Center of Smart Interfaces, Technische Universität Darmstadt, Germany*
[4] *currently at Department of Civil and Environmental Engineering, Princeton University, USA.*
*Corresponding author: volker.ebert@ptb.de*
# Abstract
Highly accurate water vapor measurements are indispensable for understanding a variety of scientific
questions as well as industrial processes. While in metrology water vapor concentrations can be defined,
generated and measured with relative uncertainties in the single percentage range, field deployable
airborne instruments deviate even under quasi-static laboratory conditions up to 10-20%. The novel
SEALDH-II hygrometer, a calibration-free, tuneable diode laser spectrometer, bridges this gap by
implementing an entirely new concept to achieve higher accuracy levels in the field. Here we present the
absolute validation of SEALDH-II at a traceable humidity generator during 23 days of permanent operation
at 15 different $H_2O$ concentration levels between 5 and 1200 ppmv. At each concentration level, we studied
the pressure dependence at 6 different gas pressures between 65 and 950 hPa. Further, we describe the
setup for this metrological validation, the challenges to overcome when assessing water vapor
measurements on a high accuracy level, as well as the comparison results. With this validation, SEALDH-II
is the first metrologically validated humidity transfer standard which links several scientific airborne and
laboratory measurement campaigns to the international metrological water vapor scale.
# 1. Introduction
Water vapor affects, like no other substance, nearly all atmospheric processes (Ludlam, 1980; Möller et al.,
2011; Ravishankara, 2012). Water vapor represents not only a large direct feedback to global warming when
forming clouds, but also plays a major role in atmospheric chemistry (Held and Soden, 2000; Houghton,
2009; Kiehl and Trenberth, 1997). Changes in the water distribution, as vapor or in condensed phases (e.g. in
clouds), have a large impact on the radiation balance of the atmosphere. This justifies that water vapor is
often mentioned as the most important greenhouse gas and one of the most important parameters in
climate research (Ludlam, 1980; Maycock et al., 2011). Water vapor is often needed for other in-situ
atmospheric analyzers to correct for their water vapor cross-interference. The high (spatial and temporal)
variability of atmospheric water vapor, its large dynamic range (3 – 40 000 ppmv), and its broad





spectroscopic fingerprint typically require complex multi-dimensional calibrations, in particular for
spectroscopic sensors. These calibrations often embrace the water vapor content of the gas flow to be
analyzed as one of the key calibration parameters even if the instrument (e.g. for $CO_2$), is not intended to
measure water vapor at all.
In particular for field weather stations, water vapor analyzers often are seen as part of the standard
instrumentation in atmospheric research. This seems reasonable due to several reasons: slow $H_2O$
concentration change over hours, the typical mid-range humidity levels (approx. above 5000 ppmv), no
significant gas pressure or temperature change, target accuracy on the order of 15%, and the absence of
"non-typical atmospheric components" such as soot or hydrophobic substances. Water vapor
measurements under these conditions can be performed by a variety of different devices (Wiederhold,
1997): Capacitive polymer sensors such as (Salasmaa and Kostamo, 1986) are frequently deployed in low
cost (field) applications. Standardized spectral absorption devices such as (Petersen et al., 2010) are often
used in research campaigns. Dew-point mirror hygrometers (DPM) are known for their high accuracy.
However, this is only true if they are regularly calibrated at a high accuracy (transfer-) standards in
specialized hygrometry laboratories such as in metrology institutes (Heinonen et al., 2012).
As soon as hygrometers have to be deployed in harsh environments (e.g. on airborne platforms), this
situation changes entirely: The ambient gas pressure (70 – 1000 hPa) and gas temperature (-80 – 40°C)
ranges are large and both values change rapidly, the required $H_2O$ measurement range is set by the ambient
atmosphere (3 – 40000 ppmv),  mechanical stress and vibrations occur, and the sampled air contains
additional substances from condensed water (ice, droplets), particles, or even aircraft fuel vapor (e.g. on
ground). These and other impacts complicate reliable, accurate, long-term stable $H_2O$ measurements and
briefly outline why water vapor measurements remain a quite difficult in-situ measurement in the field,
even if they are nearly always needed in atmospheric science. Up to now, the lack of sufficient accuracy
may have limited important scientific interpretations (Krämer et al., 2009; Peter et al., 2006; Scherer et al.,
2008; Sherwood et al., 2014).
Over the last decades, numerous hygrometers were developed and deployed on aircraft (Buck, 1985; Busen
and Buck, 1995; Cerni, 1994; Desjardins et al., 1989; Diskin et al., 2002; Durry et al., 2008; Ebert et al., 2000;
Gurlit et al., 2005; Hansford et al., 2006; Helten et al., 1998; Hunsmann et al., 2008; Karpechko et al., 2014;
Kley and Stone, 1978; May, 1998; Meyer et al., 2015; Ohtaki and Matsui, 1982; Roths and Busen, 1996;
Salasmaa and Kostamo, 1986; Schiff et al., 1994; Silver and Hovde, 1994a, 1994b; Thornberry et al., 2014;
Webster et al., 2004; Zöger et al., 1999a, 1999b) (non-exhaustive list), but those often show results which are
not sufficient for validation or falsification of atmospheric models in terms of the required absolute
accuracy, precision, temporal resolution, long-term stability, comparability, etc. These problems can be
grouped into two major categories: accuracy linked problems and time response linked problems. The latter
one is in particular important for investigations in strongly, spatially structured regions in the lower
troposphere as well as for investigations in clouds. In these regions, even two on average agreeing
instruments with different response times yield local, large, relative deviations on the order of up to 30%
(Smit et al., 2014). In contrast to time response studies, accuracy linked problems in flight are difficult to



isolate since they are always covered by the spatial variability (which leads to temporal variability for
moving aircraft) of atmospheric $H_2O$ distribution. Comparing hygrometer in flight, such as, for example in
(Rollins et al., 2014), does not facilitate a clear accuracy assessment.
Therefore in 2007, an international intercomparison exercise named "AquaVIT" (Fahey et al., 2014) was
carried out to compare airborne hygrometers under quasi-static, laboratory-like conditions for upper
tropospheric and stratospheric humidity levels. AquaVIT (Fahey et al., 2014) encompassed 22 instruments
from 17 international research groups. The instruments were categorized in well-validated, often deployed
"core" instruments (APicT, FISH, FLASH, HWV, JLH, CFH) and "younger" non-core instruments.
AquaVIT revealed in the important 1 to 150 ppmv $H_2O$ range, that -even under quasi-static conditions- the
deviation between the core instrument's readings and their averaged group mean was on the order of ±10
%. This result fits to the typical interpretation problems of flight data where instruments often deviate from
each other by up to 10%, which is not covered by the respective uncertainties of the individual instruments.
AquaVIT was a unique first step to document and improve the accuracy of airborne measurements in order
to make them more comparable. However, no instrument could claim after AquaVIT that its accuracy is
higher than any other AquaVIT instrument, since no "gold standard" was part of the campaign, i.e., a
metrological transfer standard (JCGM 2008, 2008; Joint Committee for Guides in Metrology (JCGM), 2009)
traced back to the SI units. There is no physical argument for the average being better than the measured
value of a single instrument. Instead, many arguments speak for systematic deviations of airborne
hygrometers: Most hygrometers have to be calibrated. Even for a perfect instrument, the accuracy issue is
entirely transferred to the calibration source and its gas handling system, which in this case leads to two
major concerns: First, one has to guarantee that the calibration source is accurate and stable under field
conditions, i.e., when using it before or after a flight on the ground. This can be challenging especially for
the transportation of the source with all its sensitive electronics/mechanics and for the deviating ambient
operation temperature from the ambient validation temperature (hangar vs. laboratory). Even more prone
to deviations are calibration sources installed inside of an aircraft due to changing ambient conditions such
as cabin temperature, cabin pressure, orientation angle of instrument (important, if liquids are used for
heating or cooling). Secondly, the gas stream with a highly defined amount of water vapor has to be
conveyed into the instrument. Especially for water vapor, which is a strongly polar molecule, this gas
transport can become a critical step. Changing from high to low concentrations or even just changing the
gas pressure or pipe temperature can lead to signal creep due to slow adsorption and desorption processes,
which can take long to equilibrate. In metrology, this issue is solved by a long validation/calibration time
(hours up to weeks, depending on the $H_2O$ concentration level), a generator without any connectors/fittings
(everything is welded) and piping made out of electro-polished, stainless steel to ensure that the
equilibrium is established before the actual calibration process is started. However, this calibration
approach is difficult to deploy and maintain for aircraft/field operations due to the strong atmospheric
variations in gas pressure and $H_2O$ concentrations, which usually leads to a multi-dimensional calibration
pattern ($H_2O$ concentration, gas pressure, sometimes also gas temperature) in a short amount of calibration
time (hours). Highly sensitive, frequently flown hygrometers like (Zöger et al., 1999a) are by their physical





principle, not as long-term stable as it would be necessary to take advantage of a long calibration session.
Besides the time issue to reach a $H_2O$ equilibrium between source and instrument, most calibration
principles for water vapor are influenced by further issues. A prominent example is the saturation of air in
dilution/saturation based water vapor generators: gas temperature and pressure defines the saturation level
(described e.g. by Sonntag's Equation (Rollins et al., 2014)), however, it is well-known that e.g. 100.0%
saturation is not easily achievable. This might be one of the impact factors for a systematic offset during
calibrations in the field. The metrology community solves this for high humidity levels with large, multi-
step saturation chambers which decrease the temperature step-wise to force the water vapor to condense in
every following step. These few examples of typical field-related problems show, that there is a reasonable
doubt that deviations in field situations are norm-distributed. Hence, the mean during AquaVIT might be
biased, i.e. not the correct $H_2O$ value.
The instruments by themselves might actually be more accurate than AquaVIT showed, but deficiencies of
the different calibration procedures (with their different calibration sources etc.) might mask this. To
summarize, AquaVIT documented a span of up to 20% relative deviation between the world's best airborne
hygrometers – but AquaVIT could not assess absolute deviations nor explain them, since a link to a
metrological $H_2O$ primary standard (i.e., the definition of the international water vapor scale) was missing.
Therefore, we present in this paper the first comparison of an airborne hygrometer (SEALDH-II) with a
metrological standard for the atmospheric relevant gas pressure (65 – 950 hPa) and $H_2O$ concentration
range (5 – 1200 ppmv). We will discuss the validation setup, procedure, and results. Based on this
validation, SEALDH-II is by definition the first airborne transfer standard for water vapor.

## 2. SEALDH-II

### 2.1. System description

This paper focuses on the metrological accuracy validation of the **S**elective **E**xtractive **A**irborne **L**aser **D**iode
**H**ygrometer (SEALDH-II). SEALDH-II is the airborne successor of the proof-of-concept spectrometer study
published in (Buchholz et al., 2014), which showed the possibility and the achievable accuracy level for
calibration-free dTDLAS hygrometry. The publication (Buchholz et al., 2014) demonstrates this for the
600 ppmv to 20000 ppmv range at standard ambient pressure). SEALDH-II integrates numerous different
principles, concepts, modules, and novel parts, which contribute to or enable the results shown in this
paper. SEALDH-II's high internal complexity does not allow a full, detailed discussion of the entire
instrument in this paper; for more details the reader is referred to (Buchholz et al., 2016). The following
brief description covers the most important technical aspects of the instrument from a user's point of view:

SEALDH-II is a compact (19" rack 4 U (=17.8 cm)) closed-path, absolute, directly Tunable Diode Laser
Absorption Spectroscopy (dTDLAS) hygrometer operating at 1.37 µm. With its compact dimensions and the



moderate weight (24 kg), it is well suited for space- and weight-limited airborne applications. The internal
optical measurement cell is a miniaturized White-type cell with an optical path length of 1.5 m. It is
connected to the airplane's gas inlet via an internal gas handling system comprising a temperature
exchanger, multiple temperature sensors, a flow regulator, and two gas pressure sensors.
Approximately 80 different instrument parameters are controlled, measured, or corrected by SEALDH-II at
any time to provide a holistic view on the spectrometer status. This extensive set of monitoring data ensures
reliable and well-characterized measurement data at any time. The knowledge about the instruments status
strongly facilitates metrological uncertainties calculations. SEALDH-II's calculated linear measurement
uncertainty is 4.3%, with an additional offset uncertainty of ±3 ppmv (further details in (Buchholz et al.,
2016)). The precision of SEADLH-II was determined via the Allan-variance approach and yielded 0.19
ppmv (0.17 ppmv·m·Hz-½) at 7 Hz repetition rate and an ideal precision of 0.056 ppmv (0.125 ppmv·m·Hz-
½) at 0.4 Hz. In general, SEALDH-II's time response is limited by the gas flow through the White-type
multi-pass measurement cell with a volume of 300 ccm. With the assumption of a bulk flow of 7 SLM at
200 hPa through the cell, the gas exchange time is 0.5 seconds.
SEALDH-II's measurement range covers 3 – 40000 ppmv. The calculated mixture fraction offset uncertainty
of ±3 ppmv defines the lower detection limit. This offset uncertainty by itself is entirely driven by the
capability of detecting and minimizing parasitic water vapor absorption. The concept, working principle,
and its limits are described in (Buchholz and Ebert, 2014). The upper limit of 40000 ppmv is defined by the
lowest internal instrument temperature, which has to always be higher than the dew point temperature to
avoid any internal condensation. From a spectroscopic perspective, SEALDH-II could handle
concentrations up to approx. 100000 ppmv before spectroscopic problems like saturation limit the accuracy
and increase the relative uncertainty beyond 4.3%.
**2.1. Calibration-free evaluation approach**
SEALDH-II's data treatment works differently from nearly all other published TDLAS spectrometers.
Typically, instruments are setup in a way that they measure the absorbance or a derivative measurand of
absorbance, and link it to the $H_2O$ concentration. This correlation together with a few assumptions about
long-term stability, cross interference, gas temperature dependence, gas pressure dependence is enough to
calibrate a system (Muecke et al., 1994). Contrarily, a calibration-free approach requires a fully featured
physical model describing the absorption process entirely. The following description is a brief overview; for
more details see e.g. (Buchholz et al., 2014, 2016; Ebert and Wolfrum, 1994; Schulz et al., 2007).
In a very simplified way, our physical absorption model uses the *extended* Lambert-Beer equation (Equation
1) which describes the relationship between the initial light intensity $I_0(\lambda)$ before the absorption path
(typically being in the few mW-range) and the transmitted light intensity $I(\lambda)$.
Equation 1: $I(\lambda) = E(t) + I_0(\lambda) \cdot Tr(t) \cdot exp[-S(T) \cdot g(\lambda - \lambda_0) \cdot N \cdot L]$
The parameter S(T) describes the line strength of the selected molecular transition. In SEADLH-II's case, the
spectroscopic multi-line fit takes into account 19 transition lines in the vicinity of the target line at 1370 nm



(energy levels: 110 – 211, rotation-vibrational combination band). The other parameters are the line shape
function $g(\lambda - \lambda_0)$, the absorber number density N, the optical path length L and corrections for light-type
background radiation E(t) and broadband transmission losses Tr(t).
Equation 1 can be enhanced with the ideal gas law to calculate the $H_2O$ volume mixing ratio c:
Equation 2: 
$$c = -\frac{k_B \cdot T}{S(T) \cdot L \cdot p} \int ln\left(\frac{I(v)-E(t)}{I_0(v) \cdot Tr(t)}\right)\frac{dv}{dt}dt$$

The additional variables in Equation 2 are: constant entities like the Boltzmann constant $k_B$; the optical path
length L; molecular constants like the line strength S(T) of the selected molecular transition; the dynamic
laser tuning coefficient $\frac{dv}{dt}$, which is a constant laser property; continuously measured entities such as gas
pressure (p), gas temperature (T) and photo detector signal of the transmitted light intensity I(v) as well as
the initial light intensity $I_0(v)$, which is retrieve during the evaluation process from the transmitted light
intensity I(v).
Equation 2 facilitates an evaluation of the measured spectra without any instrument calibration at any kind
of water vapor reference (Buchholz et al., 2014; Ebert and Wolfrum, 1994; Schulz et al., 2007) purely based
on first principles. Our concept of a fully calibration-free data evaluation approach (this excludes also any
referencing of the instrument to a water standard in order to correct for instrument drift, offsets,
temperature dependence, pressure dependence, etc.) is crucial for the assessment of the results described in
this publication. It should be noted that the term "calibration-free" is frequently used in different
communities with dissimilar meanings. We understand this term according to the following quote (JCGM
2008, 2008): "calibration (…) in a first step, establishes a relation between the measured values of a quantity
with measurement uncertainties provided by a measurement standard (…), in a second step, [calibration]
uses this information to establish a relation for obtaining a measurement result from an indication (of the
device to be calibrated)". Calibration-free in this sense means, that SEALDH-II does not use any
information from "calibration-, comparison-, test-, adjustment-" runs with respect to a higher accuracy
"water vapor standard" to correct or improve any response function of the instrument. SEALDH-II uses as
described in (Buchholz et al., 2016) only spectroscopic parameters and the 80 supplementary parameters as
measurement input to calculate the final $H_2O$ concentration. The fundamental difference between a
calibration approach and this stringent concept is that only effects which are part of our physical model are
taken into account for the final $H_2O$ concentration calculation. All other effects like gas pressure or
temperature dependencies, which cannot be corrected with a well-defined physical explanation, remain in
our final results even if this has the consequence of slightly uncorrected results deviations. This strict
philosophy leads to measurements which are very reliable with respect to accuracy, precision and the
instrument's over-all performance. The down-side is a relatively computer-intensive, sophisticated
evaluation. As SEALDH-II stores all the raw spectra, one could – if needed for whatever reason – also
calibrate the instrument by referencing it to a high accuracy water vapor standard and transfer the better
accuracy e.g. of a metrological standard onto the instrument. Every calibration-free instrument can be
calibrated since pre-requirements for a calibration are just a subset of the requirements for a calibration-free





instrument. However, a calibration can only improve the accuracy for the relatively short time between two
calibration-cycles by adding all uncertainty contributions linked to the calibration itself to the system. This
is unpleasant or even intolerable for certain applications and backs our decision to develop a calibration-
free instrument to enable a first principles, long-term stable, maintenance-free and autonomous hygrometer
for field use e.g. at remote sites or aircraft deployments.

## 3. SEALDH-II validation facility

### 3.1. Setup

Figure 1 right shows the validation setup. As a well-defined and highly stable $H_2O$ vapor source, we use a
commercial Thunder scientific model (TSM) 3900, similar to (Thunder-Scientific, 2016). This source
saturates pre-dried air at an elevated gas pressure in an internally ice covered chamber. The gas pressure in
the chamber and the chamber's wall temperature are precisely controlled and highly stable and thus define
the absolute water vapor concentration via the Sonntag equation (Sonntag, 1990). After passing through the
saturator, the gas expands to a pressure suitable for the subsequent hygrometer. The pressure difference
between the saturation chamber pressure and the subsequent step give this principle its name "two
pressure generator". The stable $H_2O$ concentration range of the TSM is 1 – 1300 ppmv for these specific
deployment conditions. This generator provides a stable flow of approximately 4 – 5 SLM. Roughly 0.5 SLM
are distributed to a frost/dew point hygrometer, D/FPH, (MBW 373 ) (MBW Calibration Ltd., 2010).
SEALDH-II is fed with approx. 3.5 SLM, while 0.5 SLM are fed to an outlet. This setup ensures that the dew
point mirror hygrometer (DPH)[1] operates close to the ambient pressure, where its metrological primary
calibration is valid, and that the gas flow is sufficiently high in any part of the system to avoid recirculation
of air. The vacuum pump is used to vary the gas pressure in SEALDH-II's cell with a minimized feedback
on the flow through the D/FPH and the TSM. This significantly reduces the time for achieving a stable
equilibrium after any gas pressure change in SEALDH-II's chamber. SEALDH-II's internal electronic flow
regulator limits the mass flow at higher gas pressures and gradually opens towards lower pressures
(vacuum pumps usually convey a constant volume flow i.e., the mass flow is pressure dependent). We
termed this entire setup "traceable humidity generator", THG, and will name it as such throughout the text.

### 3.1. Accuracy of THG

The humidity of the gas flow is set by the TSM generator but the absolute $H_2O$ values are traceably
determined with the dew point mirror hygrometer (D/FPH). The D/FPH, with its primary calibration, thus
guarantees the absolute accuracy in this setup. The D/FPH is not affected by the pressure changes in
SEADLH-II's measurement cell and operates at standard ambient gas pressure and gas temperature where

---

[1] The used dew point mirror hygrometer can measure far below 0°C; therefore, it is a dew point mirror above > 0°C and a frost point mirror as soon as there is ice on the mirror surface. We will use both DPH and D/FPH abbreviations interchangeably.



its calibration is most accurate. The D/FPH was calibrated (Figure 2) at the German national standard for
mid-range humidity (green, 600 – 8000 ppmv) as well as at the German national standard for low-range
humidity (blue, for lower values 0.1 – 500 ppmv). The two national standards work on different principles:
The two pressure principle (Buchholz et al., 2014) currently supplies the lower uncertainties (green, "±"-
values in Figure 2). Uncertainties are somewhat higher for the coulometric generator (Mackrodt, 2012) in
the lower humidity range (blue). The "Δ"-values in Figure 2 show the deviations between the readings of
the D/FPH and the "true" values of the national primary standards.

## 4. SEALDH-II validation procedure

### 4.1. Mid-term multi-week permanent operation of SEALDH-II

One part of the validation was a permanent operation of SEALDH-II over a time scale much longer than the
usual air or ground based scientific campaigns. In this paper, we present data from a permanent 23 day
long (550 operation hours) operation in automatic mode. Despite a very rigorous and extensive monitoring
of SEALDH-II's internal status, no malfunctions of SEALDH-II could be detected. One reason for this are
the extensive internal control and error handling mechanisms introduced in SEALDH-II, which are
mentioned above and described elsewhere (Buchholz et al., 2016). Figure 3 shows an overview of the entire
validation. The multi-week validation exercise comprises 15 different $H_2O$ concentration levels between 2
and 1200 ppmv. At each concentration level, the gas pressure was varied in six steps (from 65 to 950 hPa)
over a range which is particularly interesting for instruments on airborne platforms operating from
troposphere to lower stratosphere. Figure 3 (top) shows the comparison between SELADH-II (black line)
and the THG setup (red). Figure 3 (bottom) shows the gas pressure (blue) and the gas temperature (green)
in SEALDH-II measurement cell. The gas temperature increase in the second week was caused by a failure
of the laboratory air conditioner that led to a higher room temperature and thus higher instrument
temperature. Figure 4 shows the 200 hPa section of the validation in Figure 3. To avoid any dynamic effects
from time lags, hysteresis of the gas setup, or the instruments themselves, every measurement at a given
concentration/pressure combination lasted at least 60 min. The data from the THG (red) show that there is
nearly no feedback of a gas pressure change in SEALDH-II's measurement cell towards the D/FPH,
respectively the entire THG. The bottom subplot in Figure 4 shows the relative deviation between the THG
and SEALDH-II. This deviation is correlated to the absolute gas pressure level and can be explained by
deficiencies of the Voigt lines shape used to fit SEALDH-II's spectra (Buchholz et al., 2014)(Buchholz et al.,
2016). The Voigt profile, a convolution of Gaussian (for temperature broadening) and Lorentzian (pressure
broadening) profiles used for SEALDH-II's evaluation, does not include effects such as Dicke Narrowing,
which become significant at lower gas pressures. Neglecting these effects cause systematic, but long-term
stable and fully predictable deviations from the reference value in the range from sub percent at
atmospheric gas pressures to less than 5 % at the lowest gas pressures described here. We have chosen not
to implement any higher order line shape (HOLS) models as the spectral reference data needed are not





available at sufficient accuracy. Further, HOLS would force us to increase the number of free fitting
parameters, which would destabilize our fitting procedure, and lead to reduced accuracy/reliability (i.e.,
higher uncertainty) as well as significantly increased computational efforts. This is especially important for
flight operation where temporal $H_2O$ fluctuations (spatial fluctuations result in temporal fluctuations for a
moving device) occur with gradients up to 1000 ppmv/s.
These well understood, systematic pressure dependent deviations will be visible in each further result plot
of this paper. The impact and methods of compensation are already discussed in (Buchholz et al., 2014). The
interested reader is referred to this publication for a more detailed analysis and description.
SEALDH-II's primary target areas of operations are harsh field environments. Stability and predictability is
to be balanced with potential, extra levels of accuracy which might not be required or reliably achievable
for the intended application. Higher order line shape models are therefore deliberately traded for a stable,
reliable, and unified fitting process under all atmospheric conditions. This approach leads to systematic,
predictable deviations in the typical airborne accessible atmospheric gas pressure range (125 – 900 hPa) of
less than 3%. One has to compare these results for assessment to the non-systematic deviations of 20%
revealed during the mentioned AquaVIT comparison campaign (Fahey et al., 2014). Hence, for
field/airborne purposes, the 3% seems to be fully acceptable – especially in highly $H_2O$ structured
environments.

### 4.1. Assessment of SEALDH-II's mid-term accuracy: Dynamic effects

Besides the pressure dependence discussed above, SEALDH-II's accuracy assessment is exacerbated by the
differences in the temporal behavior between the THG's dew/frost point mirror hygrometer (D/FPH) and
SEALDH-II: Figure 5 (left) shows an enlarged 45 min. long section of measured comparison data. SEALDH-
II (black) shows a fairly large water vapor variation compared to the THG (red). The precision of SEALDH-
II (see chapter 2) is 0.056 ppmv at 0.4 Hz (which was validated at a $H_2O$ concentration of 600 ppmv
(Buchholz et al., 2016)) yielding a signal to noise ratio of 10700. Therefore, SEALDH-II can very precisely
detect variations in the $H_2O$ concentration. Contrarily, the working principle of a D/FPH requires an
equilibrated ice/dew layer on the mirror. As an indirect, inertia, thermal adjustment process, the response
time of a dew/frost point mirror hygrometer has certain limitations due to this principle (the dew/frost
point temperature measurement is eventually used to calculate the final $H_2O$ concentration), whereas the
optical measurement principle of SEALDH-II is only limited by the gas transport, i.e., the flow (exchange
rate) through the measurement cell. The effect of those different response times is clearly visible from 06:00
to 06:08 o'clock in Figure 5. The gas pressure of SEALDH-II's measurement cell (blue), which is correlated
to the gas pressure in the THG's ice chamber, shows an increase of 7 hPa – caused by the regulation cycle of
the THG's generator (internal saturation chamber gas pressure change). The response in the THG frost
point measurement (green, red) shows a significant time delay compared to SEALDH-II, which detects
changes approx. 20 seconds faster. This signal delay is also clearly visible between 06:32 to 06:40 o'clock,
where the water vapor variations detected by SEALDH-II are also visible in the smoothed signals of the



THG. Figure 5 right shows such a variation in detail (5 min). The delay between the THG and SEALDH-II is
here also approximately 20 seconds. If we assume that SEALDH-II measures (due to its high precision) the
true water vapor fluctuations, the relative deviation can be interpreted as overshooting and undershooting
of the D/FPH's controlling cycle, which is a commonly known response behavior of slow regulation
feedback loops to fast input signal changes. The different time responses lead to "artificial" noise in the
concentration differences between SEALDH-II and THG. Theoretically, one could characterize this behavior
and then try to correct/shift the data to minimize this artificial noise. However, a D/FPH is fundamentally
insufficient for a dynamic characterization of a fast response hygrometer such as SEALDH-II. Thus, the
better strategy is to keep the entire system as stable as possible and calculate mean values by using the
inherent assumption that under- and overshoots of the DPM affect the mean statically and equally. With
this assumption, the artificial noise can be seen in the first order as Gaussian distributed noise within each
pressure step (Figure 4) of at least 60 min. The error induced by this should be far smaller than the above
discussed uncertainties of the THG (and SEALDH-II).

## 5. Results

The results of this validation exercise are categorized in three sections according to the following conditions
in atmospheric regions: mid-tropospheric range: 1200 – 600 ppmv (Figure 6), upper tropospheric range: 600
– 20 ppmv (Figure 7), and lower stratospheric range: 20 – 5 ppmv (Figure 8). This categorization is also
justified by the relative influence of SEALDH-II's calculated offset uncertainty of ±3 ppmv (Buchholz and
Ebert, 2014): At 1200 ppmv, its relative contribution of 0.25% is negligible compared to the 4.3% linear
uncertainty of SEALDH-II. At 5 ppmv, the relative contribution of the offset uncertainty is 60% and thus
dominates the linear uncertainty. Before assessing the following data, it should be emphasized again that
SEALDH-II's spectroscopic first-principles evaluation was designed to rely on accurate spectral data
instead of a calibration. SEALDH-II was never calibrated or referenced to any kind of reference humidity
generator or sensor.

### 5.1. The 1200 – 600 ppmv range

Figure 6 shows the summary of the pressure dependent validations in the 1200 – 600 ppmv range. Each of
the 48 data points represents the mean over one pressure measurement section of at least 60 min (see Figure
4). A cubic polynomial curve fitted to the 600 ppmv results (blue) serves as an internal quasi-reference to
connect with the following graphs. The 600 ppmv data (grey) are generated via a supplementary
comparison at a different generator: The German national mid-humidity primary generator (PHG). This
primary generator data at 600 ppmv indicate a deviation between PHG and THG of about 0.35 %, which is
compatible with the uncertainties of the THG (see chapter 3.1) and the PHG (0.4%) (Buchholz et al., 2014).
The PHG comparison data also allow a consistency check between the absolute values of (see Figure 2) the
PHG (primary standard = calibration-free), the THG (DPM calibrated) and SEALDH-II (calibration-free).


### 5.2. The 600 – 20 ppmv range

In this range, the linear uncertainty (4.3%) and the offset uncertainty (±3 ppmv) have both a significant contribution. Figure 7 shows a clear trend: The lower the concentration, the higher the deviation. We believe this is being caused by SEALDH-II's offset variation and will be discussed in the 20 – 5 ppmv range.

### 5.3. The 20 – 5 ppmv range

The results in this range (Figure 8) are dominated by the offset uncertainty. It is important to mention at this point, that the ±3 ppmv uncertainties are calculated based on assumptions, design innovations, and several independent, synchronous measurements which are automatically done while the instrument is in operation mode (see publication (Buchholz et al., 2016; Buchholz and Ebert, 2014)). Hence, the calculated uncertainties resemble an upper uncertainty threshold; the real deviation could be lower than 3 ppmv. A clear assessment is fairly difficult since at low concentrations (i.e., low optical densities) several other effects occur together such as, e.g., optical interference effects like fringes caused by the very long coherence length of the used laser. However, Figure 9 (left) allows a rough assessment of the offset instability. This plot shows all the data below 200 ppmv, grouped by the gas pressure in the measurement cell. If one ignores the 65 hPa and 125 hPa measurements, which are clearly affected by higher order line shape effects (see above), the other measurements fit fairly well in a ±1 ppmv envelope function (grey). In other words, SEALDH-II's combined offset "fluctuations" are below 1 ppmv $H_2O$. All validation measurements done with SEALDH-II during the last years consistently demonstrated a small offset variability so that the observed offset error is around 0.6 ppmv — i.e., only 20% of the calculated ± 3 ppmv.

### 5.4. General evaluation

Figure 9 presents a summary of all 90 analyzed concentration/pressure-pairs during the 23 days of validation. The calculated uncertainties (linear 4.3% and offset ±3 ppmv) of SEALDH-II are plotted in purple. This uncertainty calculation doesn't include line shape deficiencies and is therefore only valid for a pressure range where the Voigt profile can be used to represent all major broadening effects of absorption lines (Dicke, 1953; Maddaloni et al., 2010). This is the case above 250 hPa. The results at 950, 750, 500, 250 hPa show that the maximum deviations, derived from these measurements, can be described with one single performance statement: linear +2.5%, offset -0.6 ppmv.

To prevent further interpretations, it should be noted that this result doesn't change the statement about SEALDH-II's uncertainties, since these are calculated and not based on any validation/calibration process. This is a significantly different approach: The holistic control/overview is one of the most important and essential differences between calibration-free instruments such as SEALDH-II and other classical spectroscopic instruments which rely on sensor calibration. SEALDH-II can guarantee correctness of measurement values within its uncertainties because any effect which causes deviations has to be included in the evaluation model – otherwise it is not possible to correct for it.

As mentioned before, any calibration-free instrument can be calibrated too (see e.g. (Buchholz et al., 2013)).



However by doing so, one must accept to a certain extent loss of control over the system, especially in
environments which are different from the calibration environment. For example, if a calibration was used
to remove an instrumental offset, one has to ensure that this offset is long-term stable, which is usually
quite difficult, as - shown by the example of parasitic water offsets in fiber coupled diode laser hygrometers
(Buchholz and Ebert, 2014). Another option is to choose the recalibration frequency high enough; i.e.,
minimizing the drift amplitude by minimizing the time between two calibrations. This, however, reduces
the usable measurement time and leads to considerable investment of time and money into the calibration
process. For the case of SEALDH-II, a calibration of the pressure dependence – of course tempting and easy
to do – would directly "improve" SEALDH-II's laboratory overall performance level from ±4.3% ±3 ppmv
to **±0.35% ±0.3 ppmv**. At first glance, this "accuracy" would then be an improvement by a factor of 55
compared to the mentioned results of AquaVIT (Fahey et al., 2014). However, it is extremely difficult – if
not impossible – to guarantee this performance and the validity of the calibration under harsh field
conditions; instead SEALDH-II would "suffer" from the same typical calibration associated problems in
stability and in predictability. Eventually, the calibration-free evaluation would define the trusted values
and the "improvement", achieved by the calibration, would have to be used very carefully and might
disappear eventually.

## 6. Conclusion and Outlook

The SEALDH-II instrument; a novel, compact, airborne, calibration-free hygrometer which implements a
holistic, first-principles directly tuneable diode laser absorption spectroscopy (dTDLAS) approach was
stringently validated at a traceable water vapor generator at the German national metrology institute (PTB).
The pressure dependent validation covered a $H_2O$ range from 5 to 1200 ppmv and a pressure range from
65 hPa to 950 hPa. In total, 90 different $H_2O$ concentration/pressure levels were studied within 23 days of
permanent validation experiments. Compared to other comparisons of airborne hygrometers - such as those
studied in the non-metrological AquaVIT campaign (Fahey et al., 2014), where a selection of the best "core"
instruments still showed an accuracy scatter of at least ± 10% without an absolute reference value - our
validation exercise used a traceable reference value derived from instruments directly linked to the
international water scale. This allowed a direct assessment of SEALDH-II's absolute performance with a
relative accuracy level in the sub percent range. Under these conditions, SEALDH-II showed an excellent
absolute agreement within its uncertainties which are 4.3% of the measured value plus an offset of ±3 ppmv
(valid at 1013 hPa). SEALDH-II showed at lower gas pressures - as expected - a stable, systematic, pressure
dependent offset to the traceable reference, which is caused by the line shape deficiencies of the Voigt line
shape: e.g. at 950 hPa, the systematic deviation of the calibration-free evaluated results could be described
by (linear +0.9%, offset -0.5 ppmv), while at 250 hPa the systematic deviations could be described by (linear
+2.5%, offset -0.6 ppmv). If we suppress this systematic pressure dependence, the purely statistical
deviation is described by linear scatter of ±0.35% and an offset uncertainty of ±0.3 ppmv.





Due to its extensive internal monitoring and correction infrastructure, SEALDH-II is very resilient against a
broad range of external disturbances and has an output signal temperature coefficient of only 0.026%/K,
which has already been validated earlier (Buchholz et al., 2016). Therefore, these results can be directly
transferred into harsh field environments. With this metrological, mid and upper atmosphere focused
validation presented here, we believe SEALDH-II to be the first directly deployable, metrologically
validated, airborne transfer standard for atmospheric water vapor. Having already been deployed in
several airborne and laboratory measurement campaigns, SEALDH-II thus directly links for the first time,
scientific campaign results to the international metrological water vapor scale.
*Data availability*
*The underlying data for the results shown in this paper are raw spectra (time vs. photo current), which are compressed*
*to be compatible with the instruments data storage. In the compressed state the total amount is approximately 6GB of*
*binary data. Uncompressed data size is approx. 60 GB. We are happy to share these data on request.*

*Author Contributions*
*Bernhard Buchholz and Volker Ebert conceived and designed the experiments. Bernhard Buchholz performed the*
*experiments; Bernhard Buchholz and Volker Ebert analyzed the data and wrote the paper.*

*Conflicts of Interest*
*The authors declare no conflict of interest*

*Acknowledgements:*
*Parts of this work were embedded in the EMPIR (European Metrology Program for Innovation and Research) projects*
*METEOMET- 1 and METEOMET-2. The authors want to thank Norbert Böse and Sonja Pratzler (PTB Germany)*
*for operating the German primary national water standard and the traceable humidity generator. Last but not least,*
*the authors thank James McSpiritt (Princeton University) for the various discussions about reliable sensor designs*
*and Mark Zondlo (Princeton University) for sharing his broad knowledge about atmospheric water vapor*
*measurements.*



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





**Figures:**


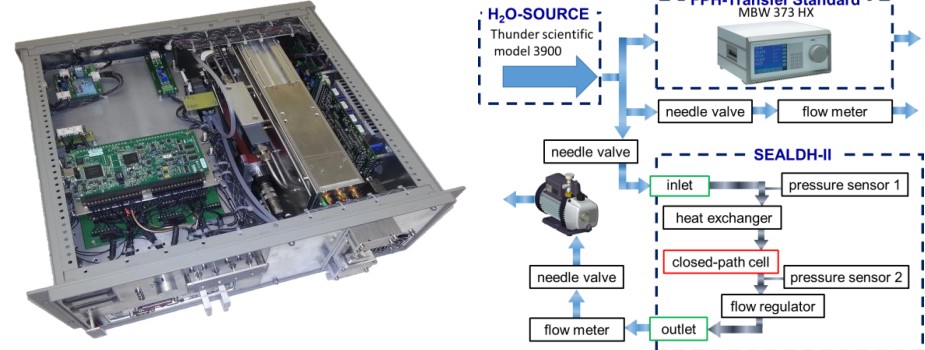

Figure 1: Left: Photo of SEALDH-II, the **S**elective **E**xtractive **A**irborne **L**aser **D**iode **H**ygrometer (dimension 19" 4 U).
Right: Setup for the metrological absolute accuracy validation. The combination of a $H_2O$ source together with a
traceable dew point hygrometer, DPM, is used as a transfer standard – a traceable humidity generator (THG).



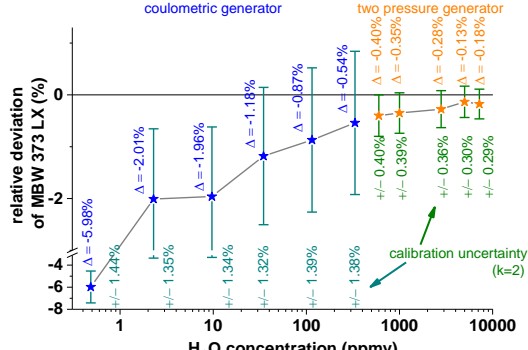

Figure 2: Calibration of the DPM (dew/frost point mirror hygrometer, MBW 373 LX, which is used as part of the THG)
at the national primary water vapor standards of Germany. The standard for the higher $H_2O$ concentration range
(orange) is a "two pressure generator" (Buchholz et al., 2014); for the lower concentration range (blue) a "coulometric
generator" (Mackrodt, 2012) is used as a reference. The deviations between reference and DPM are labelled with "Δ".
The uncertainties of every individual calibration point are stated as green numbers below every single measurement
point.








Figure 3: Overview showing all data recorded over 23 days of validation experiments. Measurements of the traceable
humidity generator (THG) are shown in red, SEALDH-II data in black, gas pressure and gas temperature in SEALDH-
II's measurement cell are shown in blue and green. Note: SEALDH-II operated the entire time without any
malfunctions; the THG didn't save data in the 35 ppmv section; the temperature increase during the 75 ppmv section
was caused by a defect of the air conditioning in the laboratory.





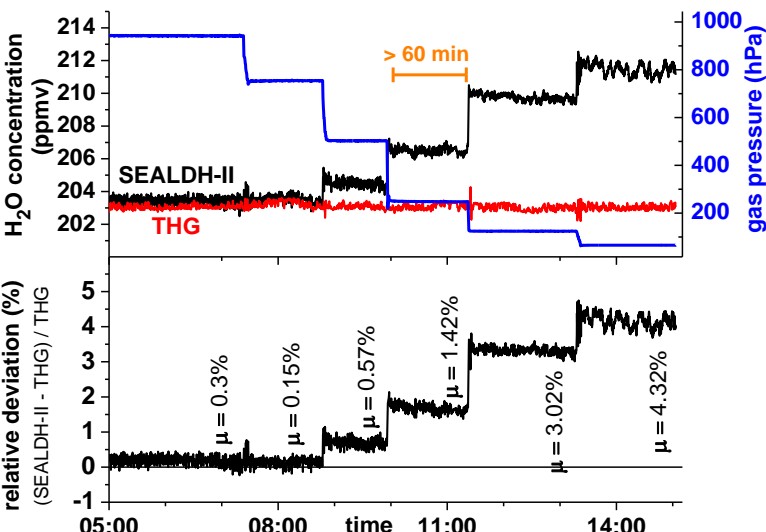

Figure 4: Detailed plot of the validation at 200 ppmv with six gas pressure steps from 50 to 950 hPa. Each individual
pressure level was maintained for at least 60 minutes in order to avoid any dynamic or hysteresis effects and to
facilitate clear accuracy assessments.



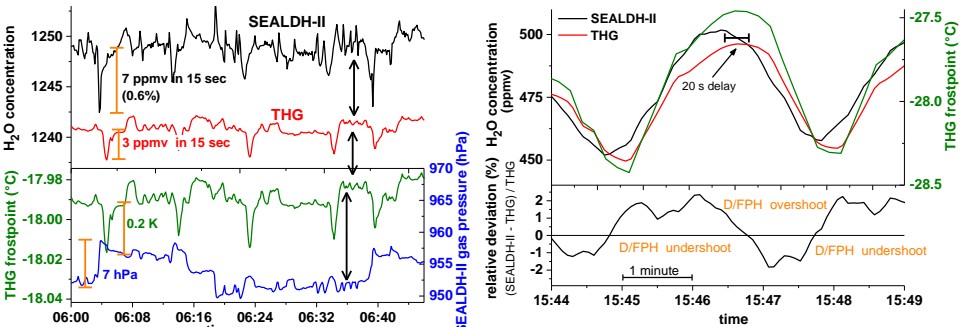

Figure 5: Short term H₂O fluctuations in the generated water vapor flow measured by SEALDH-II and the dew/frost
point mirror hygrometer (D/FPH) of the traceable humidity generator (THG). The different dynamic characteristics of
SEALDH-II (fast response time) and THG (quite slow response) lead in a direct comparison to artificial noise.






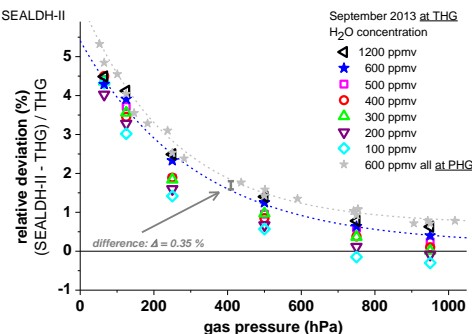

Figure 6: Gas pressure dependent comparison between SEALDH-II and THG over a $H_2O$ concentration range from 600
to 1200 ppmv and a pressure range from 50 to 950 hPa. The 600 ppmv values (in grey) are measured directly at the
national primary humidity generator (PHG) of Germany; all other $H_2O$ concentration values are measured at and
compared to the traceable humidity generator (THG). All SEALDH-II spectra were evaluated with a calibration-free
first principles evaluation based on absolute spectral parameters. No initial or repetitive calibration of SEALDH-II with
respect to any "water reference" source was used.


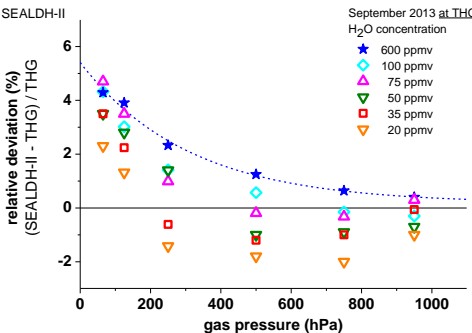

Figure 7: Comparison results as in Figure 6 but for the 200 – 600 ppmv range.


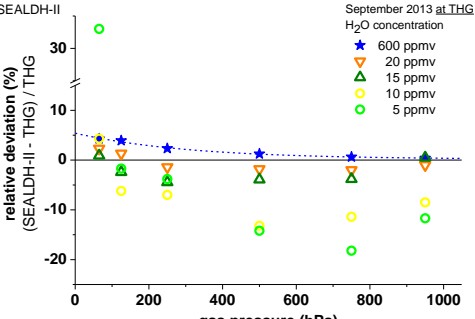




Figure 8: Comparison results as in Figure 6 and Figure 7 but for the 5 – 20 ppmv range. All spectra are determined with
a calibration-free first principles evaluation concept. The major contribution to the higher fluctuations at lower
concentrations is the accuracy of the offset determination (details see text).

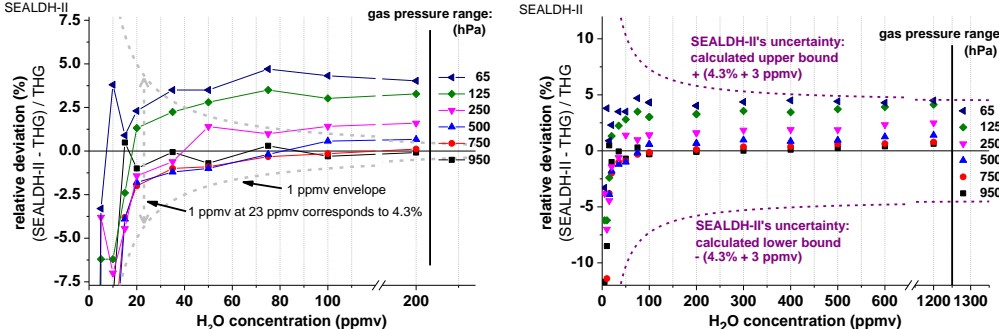

Figure 9: Direct comparison of SEALDH-II versus THG for $H_2O$ concentrations between 5 and 200 ppmv and gas

pressures from 65 to 950 hPa. Both figures show the relative deviations between SEALDH-II and THG grouped and

color-coded by gas-pressure. Left plot: relative deviations of SEALDH-II versus THG below 200 ppmv; the grey line

indicates the computed relative effect in SEALDH-II's performance caused by ±1 ppmv offset fluctuation. This line

facilitates a visual comparison between an offset impact and the 4.3% linear uncertainty of SEALDH-II. Right plot:

relative deviations for all measured data in the same concentration range. Also shown is SEALDH-II's total uncertainty

of 4.3% ±3 ppmv (calculated for 1013 hPa) as a dashed line.


