# Peer review of "Absolute, pressure dependent validation of a calibration-free, airborne laser hygrometer transfer standard (SEALDH-II) from 5 – 1200 ppmv using a metrological humidity generator"

_Atmospheric Measurement Techniques, 2016_

## Editor Comment (EC1) · D. Feist (Editor) · 23 Feb 2017

Dear authors,

thank you for adding links to the DOIs in your reference list. Please note that some of them are broken and should be fixed in the final version:

- l. 473-475: Busen et al., 1995 - DOI hyperlink does not resolve!

- l. 476-478: Cerniet et al., 1994 - DOI hyperlink does not resolve!

- l. 521-524: Hunsmann et al., 2008 - DOI hyperlink does not resolve!

- l. 525-528: JCGM 2008 - DOI hyperlink does not resolve!

- l. 534-536: Kiehl et al., 1997 - DOI hyperlink does not resolve!

- l. 584-585: Roths et al., 1996 - DOI hyperlink does not resolve!

There is no need to produce a new manuscript version at this point.

Kind regards

Dietrich Feist, AMT Associate Editor

---

## Editor Comment (EC2) · D. Feist (Editor) · 23 Feb 2017

Dear Volker Ebert,

**Thank you for your comment. I guess that is caused by the "AMT-PDF(PDF)" converter in combination with specific browser settings:**

no, it is not a browser problem. On second glance, I realized that the problems correspond only to the DOIs that contain a line break.

For example, look at Hunsman et al. (l. 521-524): If you copy and paste the full DOI (respectively the corresponding URL) from the visible text and remove the line break, you get

http://dx.doi.org/10.1007/s00340-008-3095-2

which resolves to the correct article.

However, if you click (!) on the DOI/URL, the target link points to

http://dx.doi.org/10.1007/s00340-008-524

which is broken. The reason is that the underlying hyperlink (which is activated when you click) includes the line break as well as the line number of the following line ("524") by mistake! In fact, if you click on the line number "524" – which should not have an underlying link – you are forwarded to the same broken URL. The continued "3095-2" part is dteached and also does not have an underlying link.

Simple solution: avoid line breaks in DOIs/URLs by placing long ones at the beginning of a new line. The problem might go away in the final version since there would be no line numbers.

As I said before, there is no need to fix this now. We should just make sure that the problem does not appear in the final typeset version.

Kind regards

Dietrich Feist

---

## Author Comment (AC1) · 23 Feb 2017

==> see attachment

[Figure]

Dear Dietrich Feist,

Thank you for your comment. I guess that is caused by the "AMT-PDF(PDF)" converter in combination with specific browser settings:

I tested in firefox:

- l. 473-475: Busen et al., 1995 - DOI hyperlink does not resolve!
http://dx.doi.org/10.1175/1520-0426(1995)012<0073:AHPHFA>2.0.CO;2
resolves to:
 http://journals.ametsoc.org/doi/abs/10.1175/1520-0426%281995%29012%3C0073%3AAHPHFA%3E2.0.CO%3B2

- l. 476-478: Cerniet et al., 1994 - DOI hyperlink does not resolve!
http://dx.doi.org/10.1175/1520-0426(1994)011<0445:AIHFAR>2.0.CO;2
resolves to:
http://journals.ametsoc.org/doi/abs/10.1175/1520-0426%281994%29011%3C0445%3AAIHFAR%3E2.0.CO%3B2

- l. 521-524: Hunsmann et al., 2008 - DOI hyperlink does not resolve!
http://dx.doi.org/10.1007/s00340-008-3095-2
resolves to:
http://link.springer.com/article/10.1007%2Fs00340-008-3095-2

- l. 525-528: JCGM 2008 - DOI hyperlink does not resolve!
http://dx.doi.org/10.1016/0263-2241(85)90006-5
resolves to:
http://www.sciencedirect.com/science/article/pii/0263224185900065?via%3Dihub

- l. 534-536: Kiehl et al., 1997 - DOI hyperlink does not resolve!
http://dx.doi.org/10.1175/1520-0477(1997)078<0197:EAGMEB>2.0.CO;2
resolves to:
http://journals.ametsoc.org/doi/abs/10.1175/1520-0477%281997%29078%3C0197%3AEAGMEB%3E2.0.CO%3B2

- l. 584-585: Roths et al., 1996 - DOI hyperlink does not resolve!
http://dx.doi.org/10.1016/1350-4495(95)00103-4
resolves to:
http://www.sciencedirect.com/science/article/pii/1350449595001034?via%3Dihub

It might be that some browser/acrobat reader versions cannot handle the "line-number" in front of the text. (At least that would explain it, because all of your links have a line break)

**Fig. 1.**

---

## Referee Comment (RC1) · Anonymous Referee #1 · 13 Mar 2017

This paper presents a comparison of the SEALDH-II hygrometer with the German PTB water vapor standards. The essential aspect of SEALDH-II is that it is calibration free and this validation effort closes the traceability chain with the German water vapor standard.

While I have only minor comments regarding the comparison with the German standard itself, this paper raises significant questions regarding its position within the water vapor observation community. The questions, which I outline below, need to be addressed

[Figure]

before this paper can be published. Therefore I would evaluate this paper as accept after major revisions.

General comments:

I have only minor comments regarding the technical work itself and go into detail these below. However, the larger concern is the novelty and importance as expressed in this paper.

The authors claim that SEALDH-II is a novel hygrometer, which is a calibration-free, tunable diode laser spectrometer that bridges the gap between metrological water vapor standard and field deployed hygrometers. However, last year the authors published work (Buchholz et al., 2016) on the novel Hygrometer for Airborne Investigations (HAI), which they developed in cooperation with the Research Center Jülich. The claims made in that paper read very similar than the claims made about SEALDH-II. Both are claimed to be calibration free, with some level of metrological traceability. However, the HAI paper by the same lead author is not even referenced here, which is quite odd. It is not clear what the connection is between these two instruments and which is more novel than the other. The authors should clarify the connection between these two instruments, before claiming that SEALDH-II is a novel hygrometer.

The authors also claim that this effort is the first metrologically validated humidity transfer standard. This may or may not be completely true. During the AquaVIT -II campaign this community made a dedicated effort for a metrological validation of a number of instruments. The authors collected all observations, but never released the metrological reference observations. However, they presented this work at several conferences. That work may actually be the first metrological validation of several transfer standards.

One of the drivers for the AquaVIT campaigns was the disagreement between some aircraft and balloon borne observations. Water vapor observations of less than 5 ppmv were the most important range of this disagreement. Given the uncertainty of SEALDH-II, this instrument would not contribute to this concentration range. Despite the unquestioned quality of the observations presented here, this raises the question of the importance of their results, especially given the frequent reference to AquaVIT. The authors should clearly point out, whether they have achieved a metrological validation of a field deployed instrument that can measure stratospheric water vapor (100 hPa, 5 ppmv) with an uncertainty of less than 5-10%.

Specific comments:

Line 18: I believe that the term 'bridges this gap by implementing an entirely new concept' is overselling their result. While their work is important, TDLAS technology is not new and has been around for quite a while. The authors own work on HAI show that this is not an 'entirely new concept'. Furthermore, given the relatively large metrological uncertainties at true stratospheric water vapor concentrations, I don't see, where a gap is being bridged.

Line 25, 'first metrologically validated': Aren't the AquaVIT-II and to some extent even the AquaVIT-I measurements metrologically validated?

Lines 38 and 57: The tropical tropopause is highly relevant for atmospheric water vapor and may show values of less than 1 ppmv. This lower limit is a common value for some regions and seasons.

Line 46: The target accuracy for field weather stations is certainly a lot lower than 15%. Field weather stations report relative humidity and 2%-5% accuracy (in RH) are more common requirements.

Line 50: WVSS-II instruments are another variant of TDL instruments. They are commercially available instruments but probably not standardized.

Lin 55: Currently, in situ observations of water vapor are done at least up to 10 hPa and at gas temperatures of less than -90 deg C.

Line 79: ... 'does not facilitate a clear accuracy assessment'. This study is still highly valuable and able to characterize the status of in situ observations during that campaign. The fact that large differences, seen in earlier campaigns, were not repeated there is of great value, even though there is no direct metrological connection.

Line 87-88: Differences of a factor of two stimulated AquaVIT. Disagreements of 10% were largely considered within the individual instrument uncertainties.

Line 90f: The goal of AquaVIT was to evaluate instruments under controlled conditions, not to rigorously evaluate each instrument's uncertainties. No gold standard was included since no recognized standard was available for this setup.

Line 128: Systematic differences of 20% and more were seen during AquaVIT at the lowest mixing ratios, i.e. below 3 ppmv. The authors should point out that SEALDH-II would not help addressing this concentration range.

Line 131: I doubt that this is the 'first comparison' with a metrological standard. Water vapor has been measured for a long time and a lot of validation efforts have happened, not all published. The AquaVIT-II activities, in which the authors have played an important role, is just one example.

Line, 158, 165, 345-349: The lower limit of 3 ppmv is a significant limitation, since the <10 ppmv range is essential for stratospheric observations. At 5 ppmv an uncertainty of 3 ppmv makes the measurement effectively useless for stratospheric research. This should be discussed in greater detail.

Lines 207ff: There are other calibration free instruments. HAI, published by the authors is one of them. Some of the frostpoint hygrometers, which are being used on aircraft and balloons may be considered calibration free in the same sense. Other TDL instruments are equally considered calibration free under the definition of the authors.

Line 216, 'accuracy': JCGM (2008) recommends not using this term in a quantitative sense. The authors should explain what they refer to here.

Lines 220ff: The authors point out later in the manuscript, that calibration in the strict sense improves the measurements only, if the ambient conditions can be replicated

during the calibration. They should elaborate on this topic and consolidate the various paragraphs throughout the manuscript.

Lines 264-267: Delete. These sentences contribute nothing and could be deleted.

Lines 301f, 'One has to compare ...' No, this comparison does not have to be done. The purpose of AquaVIT was very different and a metrological standard was not available at that time. This statement should be deleted.

Lines 306fff (section 4.1): Isn't the point of controlled static setups to minimize the impact of dynamic effects on the uncertainty estimation? Fundamentally the uncertainty of the SEALDH-II cannot be better than that of the THG. Therefore, the authors should quantify the impact of the THG dynamic effects on their static uncertainty estimation of SEALDH-II, if that is possible.

Line 314: What is an 'indirect, inertia, thermal adjustment process'? The authors should find a better term for what is meant here.

Line 342: PHG should be Primary Humidity Generator.

Lines 365ff, 'It is important ...': What does this sentence mean? Any uncertainty estimate always implies that the true uncertainty could be smaller. It could also be at the estimate. Lines 376 through 378 are somewhat contradictory. The authors place great value that the measurements presented here are the first metrological validation of SEALDH-II. How would non-metrological validations done previously provide contribute? Do the authors imply that non-metrological validations are equally useful or even better suited to address the uncertainty issue at low pressures and low mixing ratios? As shown in this manuscript, the uncertainty of SEALDH-II at true stratospheric values (low pressure and low mixing ratios) is too large to be scientifically relevant.

Lines 428f: Why the authors would want to suppress this systematic pressure dependence? In instrument comparisons and atmospheric measurements the systematic biases are often the determining factors.

Line 667f: What do the authors want to say here? The sentence as is doesn't make sense.

Figures 6-8: The abscissa should be shown as the Log of P. This makes it easier to relate the altitude and emphasizes the lower pressures, where water vapor is more challenging.

The authors use the term 'calibration-free' excessively and should reduce it to the necessary amount. The term is defined in a dedicated section and does not need to be repeated subsequently.

The authors do not seem to be completely familiar with the water vapor observation community. Stratospheric water vapor is also observed on large and small balloons reaching all the way into the middle stratosphere. These measurements use a variety of techniques, none of which are referenced, but should be referenced. Water vapor is also measured using remote sensing (Raman and DIAL lidar), which are technologies comparable to their own. In particular DIAL measurements are considered calibration free and traceable measurements.

Lines 61-64: These statements are much too broad and even incorrect. The vast majority of water vapor observations has been quite sufficient for validation studies of models. The limiting factor in model validation is usually the availability and coverage of these observations, not their quality. The authors should change this statement.

Technical comments:

Line 52: standard (singular)

Line 60: delete 'a quite'

Line 60: measurements (plural)

Line 73: Better: The latter is particularly important for investigations in heterogeneous regions in the lower troposphere as well as for investigations in clouds.

Line 101: . . . inside the aircraft. . .

Line 281: Replace ')(' with ', '

Line 359: Delete '(primary standard = calibration-free)', which is a meaningless repetition here. Also delete 'calibration-free' in the same line, which is again a repetition.

Line 155, 389, 413: What is the meaning of 'holistic' in this paper? Better to delete this term.

---

## Referee Comment (RC2) · Anonymous Referee #2 · 23 Mar 2017

The paper addresses the relevant scientific questions on how to measure atmospheric water vapour more accurately. The experiments are thoroughly conducted and the results well discussed, as they seem to tackle the real measurement issues. The text gives enough details and clarifications, so that it is fairly easy to follow, although it would benefit from shortening it a bit.

My main comment is about the argument that the instrument is calibration-free. The authors do discuss this in page 6, however I believe more careful wording would be

needed. Namely, the instrument does indeed measure the water vapour concentration without relating it to the quantity of the same kind (humidity). This could arguably be called an absolute measurement, where water vapour concentration is indirectly measured through quantities of different kind by using an improved physical model. However, that is in essence true also for any other instrument type, e.g. gravimetric hygrometer through mass, chilled mirror hygrometer through temperature, an impedance-based hygrometer through impedance etc. Even though the authors do fairly discuss what they mean by calibration-free, it should still be noted, that in order to obtain the water vapour concentration indirectly, the instrument has to measure different parameters directly (temperature, pressure etc.),... which eventually requires a calibration of the individual instruments.

It could be further discussed, though, weather the principle gives a potential to serve as a primary standard. They (the primary standards) do employ the absolute measurement in this sense, but they also need to be generally accepted (or chosen by convention, according to VIM). A similar situation is with chilled mirror hygrometer, which is not treated as a primary standard, but is nevertheless typically used in conjunction with it (or the SPRTs with fixed points for instance). And regularly calibrated against it.

In this respect also a more evaluation of the long-term drift would need to be conducted before a new metrological classification could be discussed, despite the argument of the offset compensation.

For this reason I would suggest to avoid the notion of calibration-free standard, but rather to stress out an alternative advantages of the SEALDTH-II and of its evaluation.

Specific comments:

- Page 1, line 23: SEALDH is not the first metrologically validated humidity standard; consider rephrasing

- Page 2, line 34: Water vapour measurement is often needed... The word measurement or similar is missing

- Page 2, line 46: consider deleting words "such as"; giving the reference is enough
Page 2, line 61: falsification is a strong word; consider revising

- Page 3, line 86: instead of "entirely transferred to", "represented by" would sound more appropriate (or similar)

- Page 3, line 100: Are you talking about desorption? If so, put it more explicitely.

- Page 4, lines 120 to 124: Please consider revising in the light of general comment above.

- Page 4, line 128: Why is it called Selective Extractive... It seems to me that Selective would be enough (selection usually means extraction).

- Page 5, line 140: Can you provide any reference for White-type cell?

- Page 5, line 146 and 147: Is the uncertainty expanded (k=2)? Please add a comment. Instead of linear uncertainty it would be better a linear part of the uncertainty or similar (the same goes for the rest of the text).

- Page 5: line 151: Authors are advised to replace units, such as ccm and SLM with the SI units through the entire paper.

- Page 5, line 161: Section 2.1 is actually 2.2. The same goes for the sections 3 and 4. Also avoid calibration-free wording.

- Page 6, line 180: variables are not constants; consider rewording... where kB is Boltzmann constant,...; S(T) is already explained in the previous page

- the second half of the page 6: please see the general comments above

- line 229: please consider replacing the word recirculation. Are you talking about back-flow due to partial pressure gradients?

- Line 234: THG seems to include both the generator and the reference instrument; I

think it's better to keep them separate (here and in the rest of the text) in order not to confuse the two purposes. Or simply use setup, where appropriate.

- Line 274 to 277: Have you considered the effect of the water vapour equations used (pure saturation pressure and the enhancement factor) at two different pressures to the deviation in response?

- Line 297 and elsewhere: precision would better be replaced by resolution

- Line 331: linear -> linear part

- Line 372: Consider replacing the . . .one single performance statement. . . with the assessment of weather the uncertainty is within the expected/estimated value.

- Line 406: water scale would better be replaced by dew-point scale or similar

- Conclusion: Please add a discussion of the long-term drift evaluation.

- Figure 4: The variable u (mi) is not expalined

---

## Author Comment (AC2) · 9 Jun 2017

We thank all the reviewers for carefully reading our manuscript and for the detailed feedback aimed at helping us to further improve the manuscript. Below we address the raised concerns in a point by point fashion. Changes are highlighted in the attached revised version.

**Anonymous Referee 1**

*This paper presents a comparison of the SEALDH-II hygrometer with the German PTB water vapor standards. The essential aspect of SEALDH-II is that it is calibration free and this validation effort closes the traceability chain with the German water vapor standard. While I have only minor comments regarding the comparison with the German standard itself, this paper raises significant questions regarding its position within the water vapor observation community. The questions, which I outline below, need to be addressed before this paper can be published. Therefore I would evaluate this paper as accept after major revisions.*

*General comments:*

*I have only minor comments regarding the technical work itself and go into detail these below. However, the larger concern is the novelty and importance as expressed in this paper. The authors claim that SEALDH-II is a novel hygrometer, which is a calibration-free, tunable diode laser spectrometer that bridges the gap between metrological water vapor standard and field deployed hygrometers. However, last year the authors published work (Buchholz et al., 2016) on the novel Hygrometer for Airborne Investigations (HAI), which they developed in cooperation with the Research Center Jülich. The claims made in that paper read very similar than the claims made about SEALDH-II. Both are claimed to be calibration free, with some level of metrological traceability. However, the HAI paper by the same lead author is not even referenced here, which is quite odd. It is not clear what the connection is between these two instruments and which is more novel than the other. The authors should clarify the connection between these two instruments, before claiming that SEALDH-II is a novel hygrometer.*

==> Thank you very much for that comment.

The instruments HAI and SEALDH-II are only similar on the first glance: The used technique (dTDLAS with a calibration-free evaluation approach) is similar, both instruments can be deployed on airborne platforms, and both are hygrometers. To grasp the differences between the instruments, we should first take look at the most important features of HAI: HAI is a multi-channel and multi-phase instrument developed for the German research aircraft HALO. The term "multi-channel" means here, that HAI measures with two different laser systems at 1.4 and 2.6 μm at two different measurement locations. The first measurement location is the "open-path" cell, which is installed on the fuselage of HALO, i.e. outside of the airplane. The second location is a "close-path" cell inside of the cabin aircraft. Additionally, HAI comprises two separate close-path cells, one for each wavelength.

The open-path cell only analyzes pure gas-phase water. By connecting the closed path cells to a forward-facing trace gas inlet, in order to sample "total water" (i.e. not only gas phase but also ice particles and droplets), the entire configuration of open and closed path cells allows "multi-phase" water measurements. In flight sections through clouds, liquid and/or frozen water particles are sampled via the inlet and then evaporated in heated sampling lines. Thus, the closed path cells determine the so called "total water". Both instruments follow on slightly different path of the concept to ensure a permanent, highly defined, multi parameter instrument control for each individual data point during the application. Goal of this is to minimize uncertainty about the validity and correctness of each measurement point. To keep the SEALDH paper concise, we maybe didn't elaborate clearly enough on this connection between the instruments.

Indeed, the novelties of HAI and SEALDH-II are in different fields.

HAI is more complex, more versatile. Its particular novelty is the multi-phase, multi-wavelength approach and its field application and validation on one of the most modern airborne carriers. HAI's unique setup allows for the first time a highly redundant overall configuration which is enabling different versions of in-flight validations e.g. of sampling errors etc. Thus, HAI is clearly the most complex and versatile, calibration-free, airborne, multi-phase hygrometer.

HAI was until now not compared and validated at a primary standard and hence does not (yet) achieve SEALDH-IIs metrological correctness/reliability/trust level. All HAI papers are more focused on field related science questions: For example, we wrote a paper about how to validate the dTDLAS relevant pressure in the open-path cell [1] which allowed us to be sure about the accuracy of HALO's gas pressure readings. However, this is not achieved for the open path gas temperature so far.

SEALDH-II on the other hand, compared to HAI, is significantly more advanced from a metrological point of view, i.e. with respect to data control, data treatment, metrological linkage, reliability level. In particular SEALDH-II is much more stringently validated at the highest primary metrological water standard. The novelty of SEALDH is thus clearly focused on the **metrological validation**.

Therefore, the full quality control from SEALDH-II still needs to be further developed, transferred and implemented in HAI. The present SEALDH manuscript deals exactly with these advantages of SEALDH II and with the results of the more stringent metrological validation experiment realized with SEALDH II. The novelty of SEALDH against HAI therefore is clearly given by the experimental rigorousness of the metrological validation. A further important novelty is that the metrological validation is realized over the full concentration and pressure range of the most relevant atmospheric regions of the UTLS and below. We added that some explanations to the manuscript, thank you for that hint!

We tried to make these differences clearer in the manuscript.

*The authors also claim that this effort is the first metrologically validated humidity transfer standard. This may or may not be completely true.*

==> We might have formulated our point not well enough, which could have led to some misconception by the reviewer. In essence, every metrological transfer standard which is used e.g. to compare different national primary standards with each other, or to transfer the representation for the metrological humidity scale to industry is a "metrologically validated humidity transfer standard. However in contrast to the reviewers statement, we wrote more detailed: *"With this validation, SEALDH-II is the first metrologically validated humidity transfer standard **which links several scientific airborne and laboratory measurement campaigns to the international metrological water vapor** scale."* We believe that this statement is true and justified, given the present manuscript and our previous papers on SEALDH. If the reviewer knows any publication about a full metrological validation of airborne laser hygrometers which contrasts or amends our statement, we would be thankful to include those in our paper and to adapt our statement.

*During the AquaVIT -II campaign this community made a dedicated effort for a metrological validation of a number of instruments. The authors collected all observations, but never released the metrological reference observations. However, they presented this work at several conferences. That work may actually be the first metrological validation of several transfer standards.*

==> It would have been very insightful if they had done exactly that. The peer reviewed paper and the White-paper [2], [3] of AquaVIT-1 explain in detail the deviations (an exemplary work!) but they are NOT linked back to any metrological source or device. The reason is obvious: At that time, the only available "metrological linked" devices were dew point mirror hygrometers (DPMH) which can be

traced back via a comparison with a primary generator. DPMH are often seen as the simple way of getting "metrological links". The DPMH used during Aquavit-1 was - to our knowledge - "only" traced back at 1 bar under static conditions. However, most of the AV-1 campaign was done at reduced pressures where the traceability was not given. What's more, it is questionable whether the traceability can be transferred from an extractive device (the DPMH) to an 80 m³ large vessel (AIDA) which was operated in a quasi-static mode, but which – in particular at the low humidity levels and gas pressures – was far from reaching a static equilibrium as it showed significant drifts over time. It is fundamentally wrong to believe, that an instrument which is used outside its validated range (pressure, temperature, concentration,…) and operation conditions for which is it built (time response, gas matrix, impurities,…) can be used as a "metrological" standard even if it was once metrologically calibrated/validated. Doint mirror hygrometers only show correct values when they are in equilibrium (ice layer, humidity, mirror temperature). At most of the AquaVIT conditions, it also has to be considered that even short (1-2 m) sampling pipes to DPMHs required equilibration times at the low humidity levels (e.g. ppmv range) of many hours up to a day in metrological institutions. Especially instruments with a low flow (e.g. DPMHs) have great problems to reach a stable equilibrium and are prone to long term drifts. Hence, the best reference instrument for AIDA would have been a sampling-free instrument like APICT, which was – however – due to many reasons not chosen as reference. We assume that the authors of the AQUAVIT papers were aware of that and therefore both papers do NOT claim any metrological linkage. Instead, all instruments were compared to the mean value of the core instruments. AquaVIT, thus answered mainly the question about the "spread between the instruments" Due to the lack of a suitable reference, the AQUAVIT comparisons are not suitable to give answers about the level of absolute accuracy. The argument which is sometimes made, that the mean value of AquaVIT is like the "true $H_2O$ value" is as wrong as to claim that one instrument was more accurate than another.

[Figure]

Referring to AquaVIT2, it is a fact that no final results and certainly no full papers have been published; neither from the simultaneous comparison exercise inside AIDA (called AV2a), which by the design of the experiment would very hardly ensure tractability, nor from the sequential metrologically motivated comparison with a PTB transfer standard generator (called AV2b). It also needs to be noted that the sequential comparison was only applied to a quite small sub group of the AV2 participants and hereby mostly to the group's internal standards and to a much lesser extend to very few instruments used in AIDA. The AV2b comparison thus had a much reduced participation compared to the AV2a part. Even if the AV2b data once will be published, they will for sure be not comparable in rigor (setup, number of data points, concentration levels, and pressure levels) and in quality (control over system, time for validation) with the data presented here. As a consequence, AV2b is of much more indirect nature, which needs up to three calibration levels/steps to link the instrument of the relevant group to the primary standard at PTB. Second, the time planned for each individual comparison in AV2b was quite short to match the total time available for AV2a. Thus, for each of the instruments/group standards only very

few data points at 1000 hPa could be measured, with a quite short measurement time per data point. The pressure dependence could not be studied in detail.

To summarize, there are no papers and certainly no peer-reviewed full papers from AV-2b or 2a. We did work on this data and published some posters on preliminary results.

*One of the drivers for the AquaVIT campaigns was the disagreement between some aircraft and balloon borne observations. Water vapor observations of less than 5 ppmv were the most important range of this disagreement. Given the uncertainty of SEALDHII, this instrument would not contribute to this concentration range. Despite the un-questioned quality of the observations presented here, this raises the question of the importance of their results, especially given the frequent reference to AquaVIT. The authors should clearly point out, whether they have achieved a metrological validation of a field de-ployed instrument that can measure stratospheric water vapor (100 hPa, 5 ppmv) with an uncertainty of less than 5-10%.*

==> Thank you for that comment; nevertheless we are also a bit puzzled by it.

SEALDH is NOT an instrument designed especially for UT/LS application, nor is it one of the common "standard" even commercially available laser hygrometers like the ones from WVSS, Picarro, LosGatos, etc. Instead, SEALDH is designed as a metrological transfer standard which is still suitable for field applications and ensures full traceability to the primary water scale. Thus, the application range is wider than just UTLS and the requirements are also much more rigid and demanding. It is clearly stated in the abstract that we claim a "primary validation", that covers the relevant mid to upper tropospheric and lower stratospheric (MT/UT/LS) concentration and pressure range. This was done via "15 different $H_2O$ concentration levels between 5 and 1200 ppmv". At each concentration level, we studied the pressure dependence at 6 different gas pressures between 65 and 950 hPa." The introduction states (as well as the referred paper in section 2 [4]) that the instrument is built for a concentration range from 3 - 40000 ppmv; 0 -1000 hPa with a calculated uncertainty of 4.3% ± 3 ppmv. The lower detection limit is defined at that point where measurement value equals uncertainty (as explained in the paper: "The calculated mixture fraction offset uncertainty of ±3 ppmv defines the lower detection limit."). The measurements in figure 8 show how conservative these estimations for the calculated uncertainties were since the "real" deviations are only in the 0 - 20% range (with the lowest level at 35%). SEALDHs science and metrology targets are thus much wider and of different perspective than those of the AQUAVIT 1 or 2.

The reasons why we explained the AquaVIT study in such a detail are the following: Firstly, it is the most extensive comparison exercise which was carried out on such a level (representative for the community, externally reviewed, blind submission, clear statements, exemplary work, etc.) that allows a reliable relative performance statement about the state of the art of airborne hygrometry. Secondly, as you can see in the graph above, AquaVIT covered the range up to 150 ppmv. Most instruments deployed during AquaVIT had a smaller concentration range compared to SEALDH-II because they were specifically developed for stratospheric measurements. With reference to the reviewer's question, we have to comment that one should not compare apples with oranges. SEALDH-II is intended as a transfer standard for the troposphere AND lower stratosphere and thus designed as a wide range hygrometer. This asks for pragmatic compromises which have consequences. If we had in mind to make a comparison **only** for the stratospheric range, we could easily replace the 1 m long extractive cell with a fiber-coupled multipath cell having 10x - 50x more path length. This would drastically improve SEALDH-II's sensitivity AND offset stability in the low concentration range. As the sensitivity and offset largely scales with the cell path length, this would lead for SEALDH-II to a calculated uncertainty (see [4]) of 4.3% ± 0.3ppmv (@ 10 m) and of 4.3% ± 0.06ppmv (@50 m) respectively, which seems quite well suited for a stratospheric application.

It also has to be kept in mind that we stated conservative calculated uncertainties; the "real" experimental deviations are usually smaller, as described and shown in this paper. In our experience – also as participant in AV 1 and 2 - we made the experience, that metrological humidity traceability options in the atmospheric sciences are more or less ignored, despite the availability of a fully validated metrological reference scale and infrastructure. It seems highly likely for us that an improved traceability would lead to better absolute accuracy of airborne hygrometry. SEALDH was designed to enable this link between metrology and field sciences. With the series of SEALDH papers and in particular with the one here under review, we intend to present the data that demonstrate this capability in a metrological sense and not just present another UTLS hygrometer.

So the argument of the reviewer that only improvements in stratospheric sensing would be required or relevant is a bit short sighted and ignores to a large extent our focus towards implementing general traceability for airborne hygrometry over the full range from LT to LS. Atmospheric science studies the entire atmospheric water cycle and not just its subsections in the stratosphere and as such, it would definitively be beneficiary to establish traceability with a wide range instrument such as SEALDH in order to improve the comparability between all atmospheric sub-compartments and not just between UTLS focused instruments.

We added a few lines to explain why we compare to AquaVIT.

*Specific comments:*
*Line 18: I believe that the term 'bridges this gap by implementing an entirely new concept' is overselling their result. While their work is important, TDLAS technology is not new and has been around for quite a while. The authors own work on HAI show that this is not an 'entirely new concept'. Furthermore, given the relatively large metrological uncertainties at true stratospheric water vapor concentrations, I don't see, where a gap is being bridged.*

==> We deleted the word "entirely" which was colloquial English.

Furthermore, the reviewer should be reminded that "The "new concept"" is not TDLAS. The new concept is validated traceability employed via a special, first principles TDLAS approach based on high-accuracy line data! It is a very common misconception that researchers who only "use" TDLAS think that all instrumental approaches have more or less identical properties – in particular with respect to the requirement for calibration. The statement to be "calibration-free" is also very often misused by lab focused TDLAS-groups without airborne experience as well as by commercial TDLAS instrument manufacturers. This is actually the reason why we term our approach "dTDLAS", in contrast to alternative evaluation approaches. Thus, overselling happens at other points of the TDLAS community, certainly not at our point. TDLAS in itself only stands for "Laser absorption spectroscopy with a tunable diode laser". "TDLAS" says nothing about the modulation approach or the data extraction and evaluation approach, so one has to be precise here.

Furthermore, SEALDH-II fortifies TDLAS with an unprecedented concept for a "complete" internal quality control mechanism, which yields a large and almost complete set of environmental/boundary parameters, the so called "housekeeping data". This ensures that every single (!) captured raw scan and hence every reported $H_2O$ value of the instrument has an assigned multi-parameter derived quality statement. This in combination with our first-principles calibration-free approach and our own high accuracy spectral data truly separates our instrument from any other, and allows us to achieve trustworthy and reliable measurements. It also ensures that a metrological validation (like the one in this work) in a protected laboratory environment can be transferred to harsh field environment (see [4] where the compensation of external effects are demonstrated with an environmental simulation chamber). It would be

helpful if the reviewer states any papers were a comparable or even better dTDLAS concept has been realized and was validated with a national primary standard, with our rigorousness.

Until then, we recommend not to generalize over an entire instrument class. But if one wants to do so, it is unavoidable to acknowledge, that instruments, which have been developed and published so far, do NOT have a data quality control concept at the level of SEALDH-II. This is often caused by the lack of a full physical model between the measurement value and the response of the instrument – the same reason why these instruments eventually have to be calibrated. The instrumental SEALDH paper [4] describes several important quality control scenarios which can be detected during SEALDH-II operation just because of the rich house-keeping data sensing any change in boundary conditions of the instrument and the measurement zone.

==> We revised the sentence.

*Line 25, 'first metrologically validated': Aren't the AquaVIT-II and to some extent even the AquaVIT-I measurements metrologically validated?*

==> As discussed above: AquaVIT was a major step towards assessing the quality of airborne hygrometers which we wrote and honored in the paper as well. Though, AquaVIT was not linked to the SI.

The Aquavit group could not / or didn't want to select a single reference instrument to serve as the main reference point, as there was obviously no knowledge about a traceability path of any of the participating instruments. As one of the participants in Aquavit 1 (and AV2), I remember well that there was no acceptance of the traceability concept and no discussion about a traceable reference instrument. We could not reach the conclusion that a single instrument was qualified as a reference type "gold standard". Hence, the best compromise the group could find was to take an average over the most mature "core" instruments. Without the concept of a unique primary reference and without any acceptance for traceability, it can't be stated that Aquavit1 is traceable, not even partially.

In Aquavit2, the only possible traceability path was planned via an extra, sequential, comparison of instruments (called AV2b) with a mobile traceable transfer standard from PTB which was transported to KIT. However, only a small number of instruments participated in the traceable comparison and no final data evaluation has been published yet. Therefore, the outcome of the traceable side-comparison called Aquavit2b is still open. Due to the large size of the AIDA vessel, it is also difficult to directly transfer the traceability to the AIDA experiment, which definitively limits the metrological impact of the Aquavit1 and 2 comparisons. Furthermore as mentioned above, AV2b was suboptimal with respect to the number of data points per instrument, the lack of comparison data at reduced pressure, and the short measurement time per data points which made it difficult to reach a stable equilibrium. Furthermore, mainly reference generators and not instruments were tested in AV2b, which results in a fairly long traceability chain over 3 and more levels, which significantly increases the uncertainty.

Note: It is unclear what the reviewer means with "to some extent"? Traceability is a binary property - so either "Yes, the instrument/data is traceable" or "No, it's not".

*Lines 38 and 57: The tropical tropopause is highly relevant for atmospheric water vapor and may show values of less than 1 ppmv. This lower limit is a common value for some regions and seasons.*

==> That's correct – even though to my knowledge averages lows are more in the range of 3-4 ppm and values below 1ppm are actually not very common. Nevertheless, we added the word "typically" to keep the statement general.

*Line 46: The target accuracy for field weather stations is certainly a lot lower than 15%. Field weather stations report relative humidity and 2%-5% accuracy (in RH) are more common requirements.*
==> That's was indeed confusing; we meant relative deviation and had more standard instrumentation in our mind rather than sophisticated instrumentation which is rarely used.
Note: Maybe we were talking about different things. We meant "relative accuracy", i.e. an absolute uncertainty of 5% rH at 50% rH or even 30% rH corresponds to a relative target uncertainty of 10% or even 16.5 %. So maybe we are not that far away from each other.
We revised the sentence.

*Line 50: WVSS-II instruments are another variant of TDL instruments. They are commercially available instruments but probably not standardized.*
==> WVSS is wavelength modulation spectroscopy based sensor which definitively has to be calibrated.
We revised the sentence

*Lin 55: Currently, in situ observations of water vapor are done at least up to 10 hPa and at gas temperatures of less than -90 deg C.*
==> Changed

*Line 79: : : : 'does not facilitate a clear accuracy assessment'. This study is still highly valuable and able to characterize the status of in situ observations during that campaign. The fact that large differences, seen in earlier campaigns, were not repeated there is of great value, even though there is no direct metrological connection.*
==> We agree with your sentence – it's not in contradiction to ours. If you look at the (Rollins et al., 2014) paper, you cannot draw a clear picture of the reason or cause for deviations: Were these deviations "real" e.g. because of different gas samples? Or: Was it caused by effects which modify the gas sample? Or by external driving forces like an increase of the instrument temperature? Environmental parasitic humidity or external pressure change? Calibration issues? Or one of the many other "airborne typical issues"?
Total comparisons like that are indeed valuable but they make it quite difficult and often even impossible to assess the internal causes of the deviation. They show more a "comparative state of the art", but make individual instrument assessments and assignments of systematic errors to certain problem or cause difficult. Hence, they do not show where improvements have to be done in the future. AquaVIT instead tried to distinguish between different effects (e.g. only the stationary, flat segments of the comparison where AIDA was as constant as possible). Deviations are only calculated within these "quasi-stationary" areas. This was done in order to suppress/minimize dynamic effects. The evaluation and comparison of the dynamic sections in AQUAVIT could still be done; to our knowledge, there is no intention to do so.

*Line 87-88: Differences of a factor of two stimulated AquaVIT. Disagreements of 10% were largely considered within the individual instrument uncertainties.*
==> Let's take a look at the AquaVIT paper: Uncertainties in the range above 5 ppmv: APicT <5%; CFH 4%; FISH-1 6%; FISH-2 6%; Flash 10%; HWV 5%; JLH 10%.
The results of AquaVIT (see plot above) show that two instruments deviate by up to 20% from each other. Keep in mind that the reference value is, as discussed above, was just a mean value – the "true" value is therefore still unknown. Even if we consider the mean value could be the "true" value, we would have the problem that the uncertainty statements of the AQUAVIT participants are insufficient. An uncertainty is NOT the typical deviation of an instrument [5], [6]. One also could rephrase the statement and state: "AquaVIT shows that the at least some instruments underestimate their uncertainty". However, I think this statement would be short-sighted since in the atmospheric communities "uncertainty", "error", "deviation", "accuracy", and "reproducibility" are often used interchangeably which causes this dilemma.

Each Aquavit participate had to state the performance or reliability or trustworthiness range of its instrument by giving an "offset term" and a relative "error". Those numbers were not analyzed nor had to be justified or even mathematically deduced and were more or less estimates by the instrument PIs. As there was no "true" reference, i.e. no validated absolute value, the error statements of the AV instrument could only be coarsely checked. Thus again, a metrological evaluation of the uncertainty by referencing an instrument to a metrological standard and to derive a metrological uncertainty is a completely different almost complementary approach.

*Line 90f: The goal of AquaVIT was to evaluate instruments under controlled conditions, not to rigorously evaluate each instrument's uncertainties. No gold standard was included since no recognized standard was available for this setup.*
==> Yes! We fully agree !
This is exactly the reason why our primary SEALDH validation is an indeed novel and highly relevant for the field! Contrary to a reviewer statement made earlier.

*Line 128: Systematic differences of 20% and more were seen during AquaVIT at the lowest mixing ratios, i.e. below 3 ppmv. The authors should point out that SEALDH-II would not help addressing this concentration range.*
==> I guess this question is referring to your third comment and hopefully already answered there. One more comment: The "span of up to 20%" describes the following graph. Keep in mind that these instruments are usually flown by themselves. If instrument A and instrument B report two measurement values, they could deviate 20% from each other (in the lower range even more).

[Figure]

*Line 131: I doubt that this is the 'first comparison' with a metrological standard. Water vapor has been measured for a long time and a lot of validation efforts have happened, not all published. The AquaVIT-II activities, in which the authors have played an important role, is just one example.*
==> The reviewer needs to be precise here: We do NOT claim the "first comparison with a metrological standard" in general, that would not be true. Instead, we present in this paper the "first comparison of an airborne hygrometer (SEALDH-II) with a metrological standard for the atmospheric relevant gas pressure (65 – 950 hPa) and H2O concentration range (5 – 1200 ppmv)". We believe that a detailed statement like this is true. In addition, we think it can be stated here, that AquaVIT-I was by no means

linked to the SI as discussed above and thus has nothing comparable to offer. For AquaVIT-II we don't understand the *diffuse, undocumented* "doubts" casted by the reviewer. We certainly don't know of any peer-reviewed final publication originating from Aquavit-2 which could challenge our claims, neither A) from the simultaneous comparison exercise in AIDA, which by the design of the experiment would be very hard to make traceable (but as said there is no full paper on AV-2), nor B) from the sequential comparison with a PTB transfer standard generator. The sequential comparison was furthermore applied only to a quite small sub group of the AV-2 participants and hereby mostly to the group's internal standards (not the instruments used in AIDA for AV-2a). This comparison part, termed AV-2b, did not have the wide participation like the AIDA part. In addition, it also did not cover the large validation range (in pressure and concentration nor did it invest sufficient amount of time as our work presented here (23 days! Permanent operation)). Even if the AV-2b data once will be published, they will for sure be not comparable in rigorousness and quality with the data presented here. One reason for this is the more indirect nature of the comparison, which needs up to three calibration levels/steps to link the instrument of the relevant groups to the primary standard at PTB. PTB just recently modified its primary standard to provide different gas pressure levels; we can for sure say that no such a comparison was linked to PTB (Germany) in the past. So again, there are no papers and certainly no peer-reviewed full papers from AV-2b. To our knowledge, there still needs to be a final data discussion meeting amongst all participants to be announced in order to decide about the further evaluation and use of the AV2 and the AV2b data.

We are of course very interested in learning which comparisons the reviewer has in mind, and would be grateful if the relevant citations could be forwarded to us. This includes publications of comparisons as well as any other metrological validations of airborne laser hygrometers, we would be very delighted to read, assess, discuss, and if suited add them.

Until then, we think our above statement is valid as well as justified and it's on the reviewer to provide references and detailed information if he is aware of comparable earlier work. To our best knowledge our data are unique in quality and in impact.

*Line, 158, 165, 345-349: The lower limit of 3 ppmv is a significant limitation, since the <10 ppmv range is essential for stratospheric observations. At 5 ppmv an uncertainty of 3 ppmv makes the measurement effectively useless for stratospheric research. This should be discussed in greater detail.*
==> We looked through the paper to make sure at any point that SEALDH-II is not presented as a stratospheric instrument as we also explained at other comments. That has never been our intention. We added some words.

*Lines 207ff: There are other calibration free instruments. HAI, published by the authors is one of them. Some of the frostpoint hygrometers, which are being used on aircraft and balloons may be considered calibration free in the same sense. Other TDL instruments are equally considered calibration free under the definition of the authors.*
==> We indeed understand the point which the reviewer wants to make, but - as explained above - we don't "oversell" and we have given in the paper presented here detailed technical explanations what we mean with "calibration-free" and even more important we have given underline{experimental justification} why we can claim this properties. It is certainly quite decisive not just to CLAIM that the instrument is calibration-free (which (too) many groups do) but also to proof it and to VALIDATE this property as we do here via a side by side comparison with the world's highest accuracy water scale and one of the few primary water standards. To our knowledge, none of the airborne TDLAS instruments has ever done this. For organizational reasons and HALO mission deployments, HAI was never available long enough in order to experimentally validate this property at the primary standard. Even though there were HAI papers coming out recently, none of them was focusing on the validation of the absolute accuracy, as

this exercise was done with the SEALDH instrument which wasn't in such long and frequent airborne missions. Again, we ask the reviewer to state papers from other air- or balloon-borne TDLAS instruments which proof a primary metrological validation exercise. We don't know any. Essentially there are many groups which claim that property but no one so far challenged, validated, and quantified experimentally the degree of accuracy which can be achieved.

Referring to a frost point instrument, we can make the following statement. Calibration-free would mean the following: The instrument is assembled and e.g. the temperature/pressure sensors (if not first principle methods) are calibrated. Then the "target" gas is guided through the frost point hygrometer and it has to report an absolute value of e.g. "500 ppmv". In an ideal world, this might be possible (using the Sonntags equation [7], ideal gas law, etc.); however, in a real world it is not. This also explains that even the most sophisticated frost/dew point hygrometer have to be calibrated regularly. Reasons for that are e.g. technical problems such as: The temperature sensor has to measure the surface (!) temperature of the mirror; the constant airflow above the mirror transfers heat away. I agreed, all this could be modelled and corrected in principle – however nobody (to our knowledge) has every published such a work. Operational problems such as: Any kind of hydrophobic substance, any dirty, micro scratches on the mirror, etc. change the ice layer behavior locally. Therefore, the "frost point" can be shifted slightly which is corrected during calibration. Physical problems: The enhancement factors (which describe the dew point shift if the gas is not ideal (Sontag equation is not accurate in this case)) cannot be calculated due to the lack of a full physical model.

*Line 216, 'accuracy': JCGM (2008) recommends not using this term in a quantitative sense. The authors should explain what they refer to here.*

==> Line 216 contains: "… philosophy leads to measurements which are very reliable with respect to accuracy, precision" and the comment is probably referred to the line: "measurement accuracy" defined as "closeness of agreement between a measured quantity and a true quantity value of a measurand. NOTE: The concept 'measurement accuracy' is not a quantity and is not given a numerical quantity value. A measurement is said to be more accurate when it offers a smaller measurement error"

As mentioned above, we believe that in non-metrological, application and field science oriented journals like AMT, we have to find a bridging language which can convey and explain the general idea and results of the scientific work. For this paper here, we decided to stay as often as possible within the terminology of the AMT community and to find a compromise between the common understood language and the rigorous metrological terminology. We strongly believe that our paper would not profit, probably be less understandable, and thus have less impact if we add not commonly used expressions such as trueness and closeness. Therefore, we assume that every reader will understand the usage of "accuracy". A metrologically much more rigorous discussion of the implications of our work will be subject to upcoming papers in common journals of the metrological community like "Metrologia" or the like.

*Lines 220ff: The authors point out later in the manuscript, that calibration in the strict sense improves the measurements only, if the ambient conditions can be replicated during the calibration. They should elaborate on this topic and consolidate the various paragraphs throughout the manuscript.*

==> Long-term stability (long = relatively longer than the drift of the instrument) of the instrument is a requirement for any calibration. We wrote: "(…) a calibration can only improve the accuracy for the relatively short time between two calibration-cycles by adding all uncertainty contributions linked to the calibration itself to the system."

This sentence referrers to every calibration process. The replication of the ambient condition is necessary if the impact of the ambient conditions on the calibrated device is unknown. As in the comment for Line

216, we will give a more rigorous discussion in planned papers for the metrological community. In the interest of keeping this manuscript in an acceptable length to content ratio we can't discuss general topics like this in the given journal and paper.

*Lines 264-267: Delete. These sentences contribute nothing and could be deleted.*
==> We don't understand why these lines should be deleted! These lines contain: "In this paper, we present data from a permanent 23 day long (550 operation hours) operation in automatic mode. Despite a very rigorous and extensive monitoring of SEALDH-II's internal status, no malfunctions of SEALDH-II could be detected. One reason for this are the extensive internal control and error handling mechanisms introduced in SEALDH-II, which are mentioned above and described elsewhere (Buchholz et al., 2016)." This comment might link to the question above "why SEALDH-II has a new concept" which we hopefully have been clarified. Typical TDLAS instruments such as e.g. the WVSS-II (mentioned above) just give a value with a minimalistic "set" of operational data. A definite assessment whether the measurement process went well or had issues inflicting the quality is not possible. SEALDH-II embodies a new, holistic, embedded concept. Therefore, this question should now be answered and the content of the sentence should be clear: "No error" with an instrument such as SEALDH-II means: "no error" and not "maybe / probably a correct measurement"

*Lines 301f, 'One has to compare : : :' No, this comparison does not have to be done. The purpose of AquaVIT was very different and a metrological standard was not available at that time. This statement should be deleted.*
==> We have already revised this sentence according to the comments above. It seems that the reviewer presumes that we do not appreciate the tremendous contribution which AquaVIT brought to the hygrometer community. We wrote (line 89) "AquaVIT was a unique first step to document and improve the accuracy of airborne measurements in order to make them more comparable". If the reviewer would like to add an even more appreciative sentence, we are happy to include it!

*Lines 306fff (section 4.1): Isn't the point of controlled static setups to minimize the impact of dynamic effects on the uncertainty estimation? Fundamentally the uncertainty of the SEALDH-II cannot be better than that of the THG. Therefore, the authors should quantify the impact of the THG dynamic effects on their static uncertainty estimation of SEALDH-II, if that is possible.*
==> This statement is correct. The figure 2 shows exactly that: The uncertainty stated there (line 643) is "The uncertainties of every individual calibration point are stated as green numbers below every single measurement point". Figure 5 left allow double checking of this statement. The max peak-to-peak deviation is approx. 4 ppmv at 1250 ppmv equals 0.3%. The Figure 5 right is an extreme case where the dew point mirror data are oscillating. A section like that is not preferable/acceptable for a validation.
We added that for clarification.
As a site note: A few comments above, the reviewer stated "2%-5% accuracy (in RH) are more common requirements". Figure 5 right shows that a measurement which has a small dynamic change can easily oscillate a frost point mirror hygrometer in the order of 2%. Therefore, a reliable accuracy of 2% is not "easily" achievable with frost point mirror hygrometers which are often seen as the most accurate hygrometers.

*Line 314: What is an 'indirect, inertia, thermal adjustment process'? The authors should find a better term for what is meant here.*
==> Revised

*Line 342: PHG should be Primary Humidity Generator.*

==> Revised, thank you

*Lines 365ff, 'It is important : : :': What does this sentence mean? Any uncertainty estimate always implies that the true uncertainty could be smaller. It could also be at the estimate. Lines 376 through 378 are somewhat contradictory. The authors place great value that the measurements presented here are the first metrological validation of SEALDH-II. How would non-metrological validations done previously provide contribute?*

==> Response to the first question: We have already answered this question to some extent above: The words "uncertainty", "error", "deviation", "accuracy", and "reproducibility" are often used interchangeably in different communities: E.g. if an instrument has a typical "deviation" of 2%, the uncertainty is sometimes just defined as 2%. From a metrological point of view, such an "uncertainty definition" has to be seen critical. An uncertainty budget/consideration should include all significant factors to understand their individual contributions. An aggregated value, such as a deviation, does not facilitate further statements about the "reliability" of this so defined uncertainty. The more the deviation measurements includes other parameters such as instrument temperature, external/internal pressure, power supply fluctuations, vibrations etc. the more reliably a deviation statement can be transformed into an uncertainty - even with a black-box-like system. SEALDH-II is designed as a fully white box system to avoid in the first place these kinds of "deviation=> uncertainty" discussions.

==> Response to the second question: We described AquaVIT as the largest, most representative, non-metrological laboratory comparison exercise. From our point of view, AquaVIT is unique in terms of quality level and the conclusions which could be drawn from it. Therefore, we did not describe other airborne or laboratory campaigns since a detailed analysis would lengthen this paper unnecessarily and would not add value since the overall statements remains. If the reviewer is interested to read more about other water vapor related comparisons: here are some entry points [8]–[15]

*Do the authors imply that non-metrological validations are equally useful or even better suited to address the uncertainty issue at low pressures and low mixing ratios? As shown in this manuscript, the uncertainty of SEALDH-II at true stratospheric values (low pressure and low mixing ratios) is too large to be scientifically relevant.*

==> First part of the question: A non-metrological validation has by definition no direct link the SI units. Therefore, it is well suited for comparing instruments relatively to each other but as soon as the absolute value is of strong interest, a metrological link should be established, to provide comparability between different instruments. This is a general recommendation and does explicitly not depend on any pressure and/or mixing ratio range.

==> Second part: SEALDH-II is – as mentioned frequently – not designed as a dedicated stratospheric instrument. It could be adopted for LS conditions but so far it isn't ; we clarified this in this rebuttal several times as well as in the paper. SEALDH-II could, as explained above, set up with a longer optical path length to serve as a standard for stratospheric values. But currently, SEALDH-II is set up for tropospheric concentrations up to 40000 ppmv and with its uncertainty of 4.3% ± 3 ppmv not well suited for single digits ppmv measurements. The measurement principle, however, could be easily adapted to LS conditions.

*Lines 428f: Why the authors would want to suppress this systematic pressure dependence? In instrument comparisons and atmospheric measurements the systematic biases are often the determining factors.*

==> We don't suppress. If we did, we would only show the "corrected" i.e. calibrated data. As we have a clear physical explanation for the pressure dependence, which is an inherent deficit of the VOIGT line

shape, we certainly know that by expansion of our physical measurement model – i.e. inclusion of a higher order line shape profile – we could "remove" this pressure deviation at the cost of higher computational efforts and an enlargement of the number of necessary fit parameters. Recent developments in higher order line shape models (such Speed-dependent-Voigt (SDV) [16] or Hartmann-Tran-Profile (HTP) [17]) are most suited for such an improvement, which we have planned for future updated versions of our fitting model.

*Line 667f: What do the authors want to say here? The sentence as is doesn't make sense.*
==> Maybe we talk about different lines: "The different dynamic characteristics of SEALDH-II (fast response time) and THG (quite slow response) lead in a direct comparison to artificial noise." We added here a sentence.

*Figures 6-8: The abscissa should be shown as the Log of P. This makes it easier to relate the altitude and emphasizes the lower pressures, where water vapor is more challenging.*
==> Thank you for that comment. SEALDH-II is a wide range (3 - 40000 ppmv) hygrometer. Therefore, the entire pressure range is equally important. This comment is probably inspired by the fact that the reviewer assumed SEALDH-II might be a purely stratospheric instrument. We clarified that (see multiple comments above)

*The authors use the term 'calibration-free' excessively and should reduce it to the necessary amount. The term is defined in a dedicated section and does not need to be repeated subsequently. The authors do not seem to be completely familiar with the water vapor observation community. Stratospheric water vapor is also observed on large and small balloons reaching all the way into the middle stratosphere. These measurements use a variety of techniques, none of which are referenced, but should be referenced. Water vapor is also measured using remote sensing (Raman and DIAL lidar), which are technologies comparable to their own. In particular DIAL measurements are considered calibration free and traceable measurements.*
==> Thank you for this comment. SEALDH-II is not a stratospheric hygrometer; therefore, we didn't focus on this single atmospheric region. The target of this paper is a "Pressure dependent absolute validation from 5 – 1200 ppmv at a metrological humidity generator" and not to write a review about different water measurement techniques even up to the middle stratosphere. As an entry point for such a review for different measurement methods: [18]
We would be interested to learn more about the "default" traceability of DIAL measurements. The technique certainly requires traceable absorption coefficients, which are very difficult to get, due to the lack of traceable spectral data. We would be delighted to receive some references about this specific topic.

*Lines 61-64: These statements are much too broad and even incorrect. The vast majority of water vapor observations has been quite sufficient for validation studies of models. The limiting factor in model validation is usually the availability and coverage of these observations, not their quality. The authors should change this statement.*
==> The statement was: "These and other impacts complicate reliable, accurate, long-term stable $H_2O$ measurements" followed by a brief outline why water vapor measurements remain a quite difficult insitu measurement in the field, even if they are nearly always needed in atmospheric science. Up to now, the lack of sufficient accuracy may have limited important scientific interpretations (Krämer et al., 2009; Peter et al., 2006; Scherer et al., 63 2008; Sherwood et al., 2014)."

==> We added the comment about availability and coverage – thank you for that.

We would like to note that we can't follow the reviewer's opinion that "quality" is not a limiting factor in atmospheric science. This is definitively too general. I remind e.g. of a paper about observations of atmospheric super saturations of 300% in the UTLS, well beyond the homogenous nucleation threshold! It is not unlikely that these were caused by significant offsets of non-traceable instruments, which then led to this extreme numbers. Other examples could open discussions on sampling effects in gas lines to extractive water sensors. Here, our 4-fold redundant HAI sensor could show in the future a quantification of such effects. We are convinced that data quality and quantification of disturbances is actually often a topic in atmospheric $H_2O$ detection.

*Technical comments:*
*Line 52: standard (singular)*
==> thank you
*Line 60: delete 'a quite'*
==> thank you
*Line 60: measurements (plural)*
==> thank you
*Line 73: Better: The latter is particularly important for investigations in heterogeneous*
*regions in the lower troposphere as well as for investigations in clouds.*
==> thank you
*Line 101: : : : inside the aircraft: : :*
==> thank you
*Line 281: Replace ')(' with ', '*
==> thank you
*Line 359: Delete '(primary standard = calibration-free)', which is a meaningless repetition*
*here. Also delete 'calibration-free' in the same line, which is again a repetition.*
==> We revised to emphasis that two calibration-free and one calibrated device are compared with each other.
*Line 155, 389, 413: What is the meaning of 'holistic' in this paper? Better to delete this*
*term.*
==> We introduces this term for a situation where a "state of the art detection principle" is embedded in a sophisticatedly controlled/supervised environment. I guess the comment above "why is SEALDH-II a new concept" and this comment are linked. We emphasized that better in the text.

[1]     B. Buchholz, A. Afchine, and V. Ebert, "Rapid, optical measurement of the atmospheric pressure on a fast research aircraft using open-path TDLAS," *Atmospheric Measurement Techniques*, vol. 7, pp. 3653–3666, (2014), doi:10.5194/amt-7-3653-2014.

[2]     D. W. Fahey, H. Saathoff, C. Schiller, V. Ebert, T. Peter, N. Amarouche, L. M. Avallone, R. Bauer, L. E. Christensen, G. Durry, C. Dyroff, R. Herman, S. Hunsmann, S. Khaykin, P. Mackrodt, J. B. Smith, N. Spelten, R. F. Troy, S. Wagner, and F. G. Wienhold, "The AquaVIT-1 intercomparison of atmospheric water vapor measurement techniques," *Atmospheric Measurement Techniques*, vol. 7, pp. 3159–3251, (2014), doi:10.5194/amtd-7-3159-2014.

[3]     D. Fahey and R. Gao, "Summary of the AquaVIT Water Vapor Intercomparison: Static Experiments," *source: https://aquavit.icg.kfa-juelich.de/WhitePaper/AquaVITWhitePaper_Final_23Oct2009_6MB.pdf (last accessed: May 2014)*, (2009).

[4]    B. Buchholz, S. Kallweit, and V. Ebert, "SEALDH-II—An Autonomous, Holistically Controlled, First Principles TDLAS Hygrometer for Field and Airborne Applications: Design–Setup– Accuracy/Stability Stress Test," *Sensors*, vol. 17, no. 1, p. 68, (2016), doi:10.3390/s17010068.

[5]    Joint Committee for Guides in Metrology (JCGM), "Evaluation of measurement data - An introduction to the 'Guide to the expression of uncertainty in measurement' and related documents," *BIPM: Bureau International des Poids et Mesures, www.bipm.org*, (2009).

[6]    JCGM 2008, "JCGM 200 : 2008 International vocabulary of metrology — Basic and general concepts and associated terms ( VIM ) Vocabulaire international de métrologie — Concepts fondamentaux et généraux et termes associés ( VIM )," *International Organization for Standardization*, vol. 3, no. Vim, p. 104, (2008), doi:10.1016/0263-2241(85)90006-5.

[7]    D. Sonntag, "Important new Values of the Physical Constants of 1968, Vapour Pressure Formulations based on the ITS-90, and Psychrometer Formulae," *Meteorologische Zeitschrift*, vol. 40, no. 5, pp. 340–344, (1990).

[8]    M. Heinonen, "A comparison of humidity standards at seven European national standards laboratories," *Metrologia*, vol. 39, no. 3, pp. 303–308, (2002), doi:10.1088/0026-1394/39/3/7.

[9]    J. M. Livingston, B. Schmid, p. b. Russell, J. Redemann, J. R. Podolske, and G. S. Diskin, "Comparison of Water Vapor Measurements by Airborne Sun Photometer and Diode Laser Hygrometer on the NASA DC-8," *Journal of Atmospheric and Oceanic Technology*, vol. 25 (10), pp. 1733–1743, (2008).

[10]   S. J. Abel, R. J. Cotton, P. a. Barrett, and a. K. Vance, "A comparison of ice water content measurement techniques on the FAAM BAe-146 aircraft," *Atmospheric Measurement Techniques Discussions*, vol. 7, no. 5, pp. 4815–4857, (2014), doi:10.5194/amtd-7-4815-2014.

[11]   M. Helten, H. G. J. Smit, and D. Kley, "In-flight comparison of MOZAIC and POLINAT water vapor measurements," *Journal of geophysical research*, vol. 104 (D21), pp. 26087–26096, (1999).

[12]   S. J. Abel, R. J. Cotton, P. A. Barrett, and A. K. Vance, "A comparison of ice water content measurement techniques on the FAAM BAe-146 aircraft," *Atmospheric Measurement Techniques*, vol. 7, no. 9, pp. 3007–3022, (2014), doi:10.5194/amt-7-3007-2014.

[13]   R. A. Ferrare, S. H. Melfi, D. N. Whiteman, K. D. Evans, F. J. Schmidlin, and D. O. Starr, "A comparison of water vapor measurements made by Raman lidar and radiosondes," *Journal of Atmospheric & Oceanic Technology*, vol. 17, pp. 1177–1195, (1995), doi:10.1175/1520-0426(1995)012<1177:ACOWVM>2.0.CO;2.

[14]   A. K. Vance, S. J. Abel, R. J. Cotton, and A. M. Woolley, "Performance of WVSS-II hygrometers on the FAAM research aircraft," *Atmospheric Measurement Techniques*, vol. 8, no. 3, pp. 1617–1625, (2015), doi:10.5194/amt-8-1617-2015.

[15]   H. Vömel, V. Yushkov, S. Khaykin, L. Korshunov, E. Kyrö, and R. Kivi, "Intercomparisons of Stratospheric Water Vapor Sensors: FLASH-B and NOAA/CMDL Frost-Point Hygrometer," *Journal of Atmospheric and Oceanic Technology*, vol. 24, no. 6, pp. 941–952, (2007), doi:10.1175/JTECH2007.1.

[16]   C. D. Boone, K. A. Walker, and P. F. Bernath, "Speed-dependent Voigt profile for water vapor in infrared remote sensing applications," *Journal of Quantitative Spectroscopy and Radiative Transfer*, vol. 105, no. 3, pp. 525–532, (2007), doi:10.1016/j.jqsrt.2006.11.015.

[17]   J. Tennyson, P. F. Bernath, A. Campargue, A. G. Cs??sz??r, L. Daumont, R. R. Gamache, J. T. Hodges, D. Lisak, O. V. Naumenko, L. S. Rothman, H. Tran, J. M. Hartmann, N. F. Zobov, J. Buldyreva, C. D. Boone, M. D. De Vizia, L. Gianfrani, R. McPheat, D. Weidmann, J. Murray, N. H. Ngo, and O. L. Polyansky, "Recommended isolated-line profile for representing high-resolution spectroscopic transitions (IUPAC technical report)," *Pure and Applied Chemistry*, vol. 86, no. 12, pp. 1931–1943, (2014), doi:10.1515/pac-2014-0208.

[18]   ManfredWendisch and J.-L. Brenguier, *Airborne Measurements for Environmental Research: Methods*

*and Instruments*. WILEY-VCH Verlag GmbH & Co. KGaA, (2013).

---

## Author Comment (AC3) · 9 Jun 2017

We thank all the reviewers for carefully reading our manuscript and for the detailed feedback aimed at helping us to further improve the manuscript. Below we address the raised concerns in a point by point fashion. Changes are highlighted in the attached revised version.

**Anonymous Referee**

*The paper addresses the relevant scientific questions on how to measure atmospheric water vapour more accurately. The experiments are thoroughly conducted and the results well discussed, as they seem to tackle the real measurement issues. The text gives enough details and clarifications, so that it is fairly easy to follow, although it would benefit from shortening it a bit.*

*My main comment is about the argument that the instrument is calibration-free. The authors do discuss this in page 6, however I believe more careful wording would be needed. Namely, the instrument does indeed measure the water vapour concentration without relating it to the quantity of the same kind (humidity). This could arguably be called an absolute measurement, where water vapour concentration is indirectly measured through quantities of different kind by using an improved physical model. However, that is in essence true also for any other instrument type, e.g. gravimetric hygrometer through mass, chilled mirror hygrometer through temperature, an impedance-based hygrometer through impedance etc. Even though the authors do fairly discuss what they mean by calibration-free, it should still be noted, that in order to obtain the water vapour concentration indirectly, the instrument has to measure different parameters directly (temperature, pressure etc.), which eventually requires a calibration of the individual instruments.*

==> Thank you very much for that comment.

In the atmospheric science communities, there are several words commonly used: Calibrated, calibration-free, self-calibration, in-situ-calibrated; outside of these communities as well: first-principles method, primary method etc.        .

In Metrology a "calibration" is defined by (JCGM 2008, 2008): "calibration (…) in a first step, establishes a relation between the measured values of a quantity with measurement uncertainties provided by a measurement standard (…), in a second step, this information is used to establish a relation for obtaining a measurement result from an indication (of the device to be calibrated)".This metrological explanation is pretty close to those calibrations "typically" used for laser- based instruments [1] and also often described in relevant individual papers such as [2]–[5].

The reason for choosing "**calibration-free**" is to unambiguously emphasize that SEALDH-II does not use / rely on any kind of such a classical calibration process based upon a water vapor reference.

The word "**self-calibration**" is used for systems which calibrate themselves during operation using an internal reference (e.g. [6]). Typical examples are temperature sensors, which use the Curie effect or phase change transitions as temperature reference points inside of the sensor. These kind of sensors do not need an *external* calibration process; however, their accuracy depends directly on the quality of the "built-in reference" and the internal calibration cycle which is done with sensor dependent strategies. "**In-situ**" or "**online**" calibrated systems follow usually measurement cycles following a pattern like "calibration – measurement – calibration etc." to cyclically link the absolute accuracy of the instrument to a reference analyzer/generator which is during the calibration process connected to the instrument to be calibrated.

From a metrological point of view, the term "calibration-free" is not as broad as the terms "first principle method" or "primary method". The word "absolute" is commonly seen as the opposite of "relative". E.g. gas pressure transmitters are sold as relative and absolute versions. If we named a calibrated instrument not as an absolute device, we would confuse many readers significantly since this view "relative to an absolute reference" seems not to be commonly used. Thus, we strongly believe that an optimum, commonly understandable word choice for our instrument characteristics which provides and ensures the desired clarity and quick comprehensibility of the idea is "calibration-free".

When we talk about "calibration-free" in the context of SEALDH-II, we do not refer to a special feature of this individual instrument realization; we refer to a feature of the entire instrument family i.e. the evaluation principle itself.

Let's think about a typical scenario of a calibrated instrument: The application requires from the instrument sufficient long-term stability (the expression "long-term" indicates here: "longer than the time span between two calibration cycles"). The correlation between measurement "value" and the real "physical quantity" has to be deterministic. Both properties together allow an absolute calibration of an instrument. Subclasses are instruments which have an offset drift; they still can be calibrated but will only provide a relative measurement rather than an absolute measurement.

Measurement principles which allow a calibration-free evaluation have in common, that they do not rely on/include any "instrumental specific" adjustment parameter and that they derive/calculate the final measurement result based on a first-principles based approach which relies on a physical model of the instrument and the measurement process. In our case, the instrumental response relies on a physical model of the light absorption induced by a spectrally sufficiently resolved ro-vibrational absorption transition of the water molecule. One of the most essential parameters is hence the line strength of the chosen water transition. This is a "molecular property", which does not have a spatial or temporal variation, which could modify the instrument response. The molecular parameters of the water transitions are in that sense ideal "transfer standards" to link instrument and measurements to the SI

The reviewer made the argument, that we still use calibrated devices inside of SEALDH-II. This is true, but these calibrations are only needed because we need absolute values for L, p, T, etc. These calibrations are not used to remove non-understood instrumental deviations/drifts outside the physical model of the measurement process, i.e. to calibrate any instrumental specific deviations. As an example: SEALDH-II needs the length of the optical path. If one wanted to get rid of this "measurement", we could do it directly: Length is defined by the speed of light and time. Time is defined by a molecular transition (energy). It would be highly impractical to set up an atomic clock inside of SEALDH-II, but it's not impossible, just impractical for the desired airborne application. In general, other national primary standard also do not rely on a complete set of "only primary principles", even if this is possible; it would be confusing to say "the primary standard is calibrated". Therefore, we believe people outside of Metrology will understand "traceable, based on first principle" as "calibration-free" and vise versa.

Different example: Let's think for a moment how our general approach contrasts the sometimes made argument that frost/dew point mirror hygrometer didn't need a calibration: Here, "calibration-free" would mean the following:   After mechanically and electronically setting up an instrument, the "target" gas is guided through the frost point hygrometer and it would report an absolute value of e.g. "500 ppmv". In an ideal world, this might be possible (using the Sonntags equation [7], ideal gas law, etc.); however, in a real world it is not. This also explains that even the most sophisticated frost/dew point hygrometer have to be calibrated regularly.

Reasons for that are e.g.:

Technical problems such as: The temperature sensor has to measure the surface (!) temperature of the mirror; the constant airflow above the mirror induces heat transfer losses; or the contamination of the condensation mirror with other gas constituents induces offsets by shifting the ice/dew formation; or the temperature sensor used for the mirror surface temperature suffers drifts caused by aging processes. We agree that all this could be modelled and corrected in principle – however nobody (to our knowledge) has ever published such a work. The model would have to be so general, that exact knowledge of the geometry (= length) of the chamber would be sufficient to calculate the correction function based on first principles.

Operational problems such as: Any kind of hydrophobic substance, any dirt, micro scratches on the mirror surface, etc. locally modify the ice/dew deposition behavior. The "frost/dew point" can be shifted slightly from the ideal situation which is corrected for via a calibration process. Therefore, the calibration-free dew point mirror hygrometer would need a broad set of additional measurement devices to analyze, detect, and quantify such influencing effects and to adopt the right frost/dew model.

Further, physical problems need to be solved: The enhancement factors (which relate the dew point shift if the gas is not ideal (Sontag equation is not accurate in this case)) cannot be calculated due to the lack of a full physical model. Condensation/freezing models are currently not entirely isolated from material properties. There seems to be no fundamental problem but the current models cannot cope with the complexity at the required very high accuracy of a research type frost point mirror. Vice versa, if dew/frost point mirrors were operated in a "relaxed" sensitivity regime, we would expect, that the small shifts induced by the problems mentioned above become insignificant, hence could be ignored, thus making a "reduced"-sensitivity dew point hygrometer also potentially calibration- free. The question what calibration-free means and how it is achieved therefore always has to be answered in the frame work of the performance requirements for a the desired application: Is the physical model of the measurement process and the realization of the corresponding instrument understood well enough so that the measurement process can be evaluated accurately enough without relying on a comparison with an external reference process (i.e. via a calibration to a reference), then a physical model could allow a calibration-free first-principles evaluation. If the accuracy requirements are too high for a given instrument/evaluation configuration then the instrument will require calibration via a comparison with an external/internal reference.

The design process of SEALDH-II was exactly governed to answer these questions: I.e. the question to be answered was e.g. what "enabling" accuracies of the spectral parameters, the temperature, pressure and length measurements and other controlling critical effects are needed and have to be embodied in the instrument in order to be able to realize a first principles TDLAS based measurement of the water vapor mixture fraction with a total accuracy sufficient for the airborne applications.

To summarize: Papers should have "clear" and commonly understandable statements for the target communities, which are here the applied atmospheric sciences. From this point of view, "calibration-free" is more widely comprehensible. "Traceable via a first principle approach" would be more precise but is much less common and understandable outside of the Metrology community, since the exact definitions of "traceability" and "first principle" is still to be disseminated more commonly.

*It could be further discussed, though, weather the principle gives a potential to serve as a primary standard. They (the primary standards) do employ the absolute measurement in this sense, but they also need to be generally accepted (or chosen by convention, according to VIM). A similar situation is with chilled mirror hygrometer, which is not treated as a primary standard, but is nevertheless typically used in conjunction with it (or the SPRTs with fixed points for instance). And regularly calibrated against it.*

==> We also thought about that, but eventually decided not to discuss that in an atmospheric sciences journal as this would be out of the scope of AMT. A discussion like this would fit better in "Metrologia" or similar more metrologically oriented journals. Due to this, we intend to provide a more metrologically oriented discussion of our results in a future paper.

The way from a "traceable via first principle" to a "national primary standard" is long-term process. Since the readers of AMT are dominated by the atmospheric sciences communities, we decided against an out-of-scope discussion focusing on general Metrology. Therefore, we also didn't explain e.g. if and how SEALDH-II could get a CMC entry.

*In this respect also a more evaluation of the long-term drift would need to be conducted before a new metrological classification could be discussed, despite the argument of the offset compensation.*

==> We fully agree with this statement. But this "long-term" evaluation needs time i.e. several months to 1-2 years to be long-term. Data in this paper show already, that the long-term stability seems higher than the uncertainty of the dew/frost point hygrometer. Therefore, a comparison like that has to be directly done at the national primary standard, which is pretty occupied and busy for service calibrations at PTB. But: We are currently working on that; e.g. the primary standard was recently upgraded with a setup to facilitate validations at different gas pressures.

*For this reason I would suggest to avoid the notion of calibration-free standard, but rather to stress out an alternative advantages of the SEALDTH-II and of its evaluation.*

==> We hopefully convinced the reviewer, that the word choice is a good compromise between metrology and atmospheric science and that it is the best fit for the atmospheric community, which is the one we are focusing on with this paper.

*Specific comments:*
*- Page 1, line 23: SEALDH is not the first metrologically validated humidity standard;*
*consider rephrasing*

==> We wrote "With this validation, SEALDH-II is the first metrologically validated humidity transfer standard which links several scientific airborne and laboratory measurement campaigns to the international metrological water vapor scale"

It emphasizes on the linkage between airborne, laboratory and Metrology. We added words to make it even clearer. To our knowledge, SEALDH is the first "airborne, metrologically validated humidity transfer standard". If there are previous publications which demonstrating these properties, we ask the reviewer to please provide a reference to the publication.

*- Page 2, line 34: Water vapour measurement is often needed: : : The word measurement or similar is missing*
==> Revised
*- Page 2, line 46: consider deleting words "such as"; giving the reference is enough*
==> Revised
*Page 2, line 61: falsification is a strong word; consider revising*
==> Revised
*- Page 3, line 86: instead of "entirely transferred to", "represented by" would sound more appropriate (or similar)*
==> Revised
*- Page 3, line 100: Are you talking about desorption? If so, put it more explicitly.*
==> We mentioned that in the sentence before: "…can lead to signal creep due to slow adsorption and desorption processes,…"
*- Page 4, lines 120 to 124: Please consider revising in the light of general comment above.*
==> We revised the last sentence

*- Page 4, line 128: Why is it called Selective Extractive: : : It seems to me that Selective*
*would be enough (selection usually means extraction).*

==> SEALDH refers to the entire instrument family. "Selective" stands here for the fact that SEALDH-II is gas species selective, i.e. we are refereeing to a very high chemical selectivity as a consequence of the high spectral resolution of the employed diode lasers. During the evaluation processing, we can distinguish between water vapor and its other phases as well as other species such as $CO_2$, $CH_4$, aircraft fuel vapor, etc. which might be in the sampled air. (Btw, this is also a major difference between a "single information based" evaluation (ice layer on frost point mirror) and a full model based spectroscopic evaluation. (See for details about SEALDH-II's evaluation [7])

"Extractive" stands for "taking" = extracting a gas sample in order to analyze it "inside" the instrument. In contrast to "open path"- TDLAS, where the gas sample remains where it is (i.e. in the atmosphere) and the light, used for sensing, is "brought" to the gas sample.

*- Page 5, line 140: Can you provide any reference for White-type cell?*

==> Sure, we published the full "technical" description in [7]. The general reference is [8]; our specialized version is: [9]. We added both references in the paper as well.

*- Page 5, line 146 and 147: Is the uncertainty expanded (k=2)? Please add a comment.*
*Instead of linear uncertainty it would be better a linear part of the uncertainty or similar*
*(the same goes for the rest of the text).*

==> The answer is yes; details in [7]. We have had the experience in other papers that readers outside of Metrology got confused since they are not used to different uncertainty models and confidence intervals. In order to avoid this confusion (and discussions with non-metrology reviewers), we deliberately adopted our performance parameters to coincide with the common positions in the atmospheric sciences community. A reviewer from a different paper commented on a similar topic that if one relies on HITRAN, he should not use metrological confidential intervals since the entire HITRAN database does not have common metrological standards for uncertainty calculations (HITRAN only deals with coarse uncertainty "classes" or "corridors"). Therefore, he told us that a "(k=2) note" would claim a knowledge which we did not have.

=> Thank you; we revised the wording

*- Page 5: line 151: Authors are advised to replace units, such as ccm and SLM with*
*the SI units through the entire paper.*

==> We changed ccm to ml. (For all mass flow calculations, it is important to distinguish between "liter per minute" and "standard liter per minute "(normalized to standard pressure))

*- Page 5, line 161: Section 2.1 is actually 2.2. The same goes for the sections 3 and 4.*
*Also avoid calibration-free wording.*

==> see above

*- Page 6, line 180: variables are not constants; consider rewording: : : where kB is*
*Boltzmann constant,: : :; S(T) is already explained in the previous page*
*- the second half of the page 6: please see the general comments above*

==> Good point; we revised.

*- line 229: please consider replacing the word recirculation. Are you talking about*
*back-flow due to partial pressure gradients?*

==> Recirculation is defined by local flow field gradients which point more than 90° away from the main flow direction; usually caused by vortex structures. Even if the main flow is forward facing, parts of the flow can go backwards. In general: The higher the flow, the lower the proportional backward facing flow.

*- Line 234: THG seems to include both the generator and the reference instrument; I think it's better to keep them separate (here and in the rest of the text) in order not to*

*confuse the two purposes. Or simply use setup, where appropriate.*

==> Yes that is true: In line 257, we introduced the abbreviation: "We termed this entire setup 'traceable humidity generator', THG, and will name it as such throughout the text." Issues which are linked to the entire setup are addressed by "THG", others by the individual components. We fully agree, that the overuse of abbreviations is tempting, however in this case, it should help to distinguish between the different parts rather than writing: "… the setup, consisting of frost point mirror, pressure control unit, water vapor source, heat control, co-flow unit, does the following…"

*- Line 274 to 277: Have you considered the effect of the water vapour equations used (pure saturation pressure and the enhancement factor) at two different pressures to the deviation in response?*

==> Short answer is "yes". The uncertainty of the national primary standard is strongly affected by the uncertainty of the Sonntag's equation as well as the factors named above. The uncertainty of the dew point mirror hygrometer (see figure 2) includes all of these considerations as well. We briefly documented the working principle of the primary standard elsewhere [10]; this publication also contains several references about the principle and the different issues when operating it.

*- Line 297 and elsewhere: precision would better be replaced by resolution*

==> The spectroscopic community defines precision differently than other communities. We used the term accordantly to [11], [12] which seems to be the community standard.

*- Line 331: linear -> linear part*

==> Revised

*- Line 372: Consider replacing the : : :one single performance statement: : : with the assessment of weather the uncertainty is within the expected/estimated value.*

==> Indeed, this sentence could be confusing. We revised it.

*- Line 406: water scale would better be replaced by dew-point scale or similar*

==> Revised

*- Conclusion: Please add a discussion of the long-term drift evaluation.*

==> We would like to do that but the measurements in this paper do not allow a reliable statement of the long-term drift. The data suggest that the long-term drift seems to be very small but this dataset does not allow quantifying it on a high accuracy level.

*- Figure 4: The variable u (mi) is not explained*

==> Thank you, we added that.

[1]     R. J. Muecke, B. Scheumann, F. Slemr, and P. W. Werle, "Calibration procedures for tunable diode laser spectrometers," *Proc. SPIE 2112, Tunable Diode Laser Spectroscopy, Lidar, and DIAL Techniques for Environmental and Industrial Measurement*, vol. 2112, pp. 87–98, (1994), doi:10.1117/12.177289.

[2]     M. Helten, H. G. J. Smit, W. Sträter, D. Kley, P. Nedelec, M. Zöger, and R. Busen, "Calibration and performance of automatic compact instrumentation for the measurement of relative humidity from passenger aircraft," *Journal of Geophysical Research: Atmospheres*, vol. 103, no. D19, pp. 25643–25652, (1998), doi:10.1029/98JD00536.

[3]     J. Podolske, G. Sachse, and G. Diskin, "Calibration and data retrieval algorithms for the NASA Langley/Ames Diode Laser Hygrometer for the NASA Transport and Chemical Evolution Over the Pacific (TRACE-P) mission," *Journal of Geophysical Research*, vol. 108, no. D20, p. 8792, (2003), doi:10.1029/2002JD003156.

[4]     D. Tátrai, Z. Bozóki, H. Smit, C. Rolf, N. Spelten, M. Krämer,  a. Filges, C. Gerbig, G. Gulyás, and G. Szabó, "Dual-channel photoacoustic hygrometer for airborne measurements: background, calibration, laboratory and in-flight intercomparison tests," *Atmospheric Measurement Techniques*, vol. 8, no. 1, pp. 33–42, (2015), doi:10.5194/amt-8-33-2015.

[5]     M. Zöger, A. Afchine, N. Eicke, M.-T. Gerhards, E. Klein, D. S. McKenna, U. Mörschel, U. Schmidt,

V. Tan, F. Tuitjer, T. Woyke, and C. Schiller, "Fast in situ stratospheric hygrometers: A new family of balloon-borne and airborne Lyman photofragment fluorescence hygrometers," *Journal of Geophysical Research*, vol. 104, no. D1, pp. 1807–1816, (1999), doi:10.1029/1998JD100025.

[6] "What Are the Differences between Self-Calibration and External Calibration?" [Online]. Available: http://digital.ni.com/public.nsf/allkb/305DB655C3C5C7628625710A005198F7.

[7] B. Buchholz, S. Kallweit, and V. Ebert, "SEALDH-II—An Autonomous, Holistically Controlled, First Principles TDLAS Hygrometer for Field and Airborne Applications: Design–Setup– Accuracy/Stability Stress Test," *Sensors*, vol. 17, no. 1, p. 68, (2016), doi:10.3390/s17010068.

[8] J. White, "Very long optical paths in air," *Journal of the Optical Society of America (1917-1983)*, vol. 66, no. 5, pp. 411–416, (1976), doi:10.1364/JOSA.66.000411.

[9] B. Kühnreich, M. Höh, S. Wagner, and V. Ebert, "Direct single-mode fibre-coupled miniature White cell for laser absorption spectroscopy," *Review of Scientific Instruments*, vol. 87, no. 2, pp. 0– 8, (2016), doi:10.1063/1.4941748.

[10] B. Buchholz, N. Böse, and V. Ebert, "Absolute validation of a diode laser hygrometer via intercomparison with the German national primary water vapor standard," *Applied Physics B*, vol. 116, no. 4, pp. 883–899, (2014), doi:10.1007/s00340-014-5775-4.

[11] D. W. Allan, "Statistics of atomic frequency standards," *Proceedings of the IEEE*, vol. 54, no. 2, pp. 221–230, (1966), doi:10.1109/PROC.1966.4634.

[12] P. Werle, R. Mücke, and F. Slemr, "The limits of signal averaging in atmospheric trace-gas monitoring by tunable diode-laser absorption spectroscopy (TDLAS)," *Applied Physics B*, vol. 57, no. 2, pp. 131–139, (1993), doi:10.1007/BF00425997.

---

## Author Response (AR2)

We thank both reviewers for carefully reading our manuscript and for the detailed feedback aimed at helping us to further improve the manuscript.

**Anonymous Referee 2**

*The discussion as a result of the two reviews reconfirmed my impression that there is very little to comment on the technical work of this manuscript. This part is very detail oriented and shows that the authors built and properly validated a very high quality hygrometer.*

We really appreciate the comments in the first review which made our paper clearer and easier to understand for the community. Currently, the remaining questions seem to focus on the two areas a) is SEALDH-II a stratospheric instrument b) how much technical details should be in an instrument validation/comparison paper.

We hope that we can clarify these remaining issues with our response, so that the second reviewer agrees with the "publish as is" recommendation of the first reviewer.

Below we address the raised concerns in a point by point fashion. Changes are highlighted in the attached revised version.

*However, my concerns regarding the novelty and importance have been reconfirmed. I particular, the connection with the earlier Hygrometer for Airborne Investigations (HAI) has not been adequately described. Furthermore, there apparently was an earlier version, SEALDH-I, which had the same features as SEALDH-II. In their replies the authors stated that "Maybe we didn't elaborate clearly enough on the connection between the instruments". That is somewhat of an understatement, since neither of the other two instruments were even mentioned. This was quite surprising, since both instruments come from the authors themselves. It is rare to see that authors ignore their own work. The authors also confirm my suspicion that these instruments are borne out of the same concept, I quote, "to ensure a permanent, highly defined, multi parameter instrument control for each individual data point". Therefore, the authors' claim that SEALDH-II is a novel concept is simply not correct and should not be described as such. This concept has already been implemented in HAI as atmospheric instrument and SEALDH-I as proof of concept. I fully appreciate the differences between HAI and SEALDH-II and the differing motivations for both instruments. HAI is more appropriate for atmospheric observations, while SEALDH-II was built with metrological traceability in mind. Nevertheless, these instruments are based on the same concept, measure the same atmospheric parameter using the same measurement principle, claiming to be calibration-free. Therefore, statement such as, I quote, "implementing a new holistic concept to achieve higher accuracy levels", made in the abstract, are simply not correct. The authors may claim novelty for HAI or possibly SEALDH-I, not for SEALDH-II. Monitoring many more instrument parameters than in HAI may improve the confidence in the observations and may allow a better characterization of the uncertainty budget. Their instrument is certainly of very high quality, but the authors cannot claim a 'new holistic' concept, since that has already been done. Instead the authors should include an extensive discussion of the relation between these instruments in the manuscript, not just the replies to the authors.*
*A key part of this manuscript is the metrological validation of SEALDH-II. Given that HAI is much*

*more suited to atmospheric measurements, the authors should elaborate, why they chose to build an entirely new instrument, instead of validating the first instrument they built for atmospheric measurements. Some of the discussion in their replies should be included into the manuscript as well as their plans for the validation of HAI.*

Thank you very much for that comment. There seem to be a sticky idea, which the reviewer enjoys discussing. However, the questions are far away from the focus of this paper:

The title of the publication is:
"SEALDH-II – a calibration-free transfer standard for airborne water vapor measurements: **Pressure dependent absolute validation from 5 – 1200 ppmv at a metrological humidity generator**"

The pure focus on this paper (which is also stressed out in the publication) is metrological validation of an airborne hygrometer: More specifically about the **first metrological pressure dependent, absolute validation in the range 5 – 1200 ppmV**.
The reviewer is much too focused on the hardware novelties. Even though we indeed wanted to mention our new holistic data concept, we did not describe it in detail or put technical details in the focus of the paper (as it has already been described elsewhere). The same also holds for the hardware of SEALDH-II (also has been described previously in more detail), which is also not in the focus of this paper. However, some coarse description of the hardware and data management is needed in order to link the contribution of these developments to the performance of the instrument. Thus, the **novelty of this manuscript comes from the metrological validation of our sensor.** Such a validation has not been done before - neither with this metrological rigor nor in this concentration range. The reviewer also didn't name us papers where something quite similar was described. Hence, the purpose of this publication is by no means the announcement of a "new" instrument or some "new" hardware, but the description and discussion of the validation experiment. This is of high relevance if instruments like SEALDH-II are to be used as mobile metrological transfer standards for atmospheric humidity measurements in the future. The need for such transfer standards can be clearly motivated from the results of the AquaVIT comparison.

The problem with the reviewer comments are therefore that they request to convert the paper more towards more hardware description and going very deeply into technical details between the different instrument generations. We are certainly very happy to discuss such details with the reviewer - might it be in an entirely technical discussion paper or in an offline conversation. But, for the presented manuscript, we have to object to the reviewers request as these details have been mentioned previously and so that there is no need for duplicating them in this manuscript. So this paper is particularly **NOT focusing** on the following topics:

a) A detailed instrument description of SEALDH-II. We have published a technical instrument description already. We believe that leaving out all these fine technical details on the hardware and the holistic data management significantly improves the clarity of the manuscript. When we pointed out some of SEALDH-II features, then we felt that these are vital for the understanding of the presented data. Further, our presented measurements are meant as a validation (for SEALDH-II to be a transfer standard) rather than an instrument calibration.
We addressed as many of reviewer questions in the first rebuttal as possible, because we understand the reviewers curiosity, but most of the questions where not related to the validation core of this paper at all. Thus, at some level we had to decide to stay focused in the interest of the readers and not include all technical details in this manuscript. In particular as they have already been mentioned elsewhere. Therefore, we wrote "Maybe we didn't elaborate clearly enough on the connection between the instruments" to answer the reviewer's question.

b) Since we focus on a "metrological, pressure dependent, absolute validation" of SEALDH-II, any detailed discussions of HAI topics are completely out of the context. HAI is, as we described extensively in the last rebuttal, a multi-phase hygrometer. The challenges for the HAI instrument measuring at two different wavelengths, and with an open-path cell are entirely different. Yes, the HAI and SEALDH developments have overlapping areas, but this is not a redundancy but absolutely beneficial, as the "metrological, pressure dependent, absolute validation" exercise done with SEALDH-II can be transferred quite well to the extractive channels of HAI! Maybe the reviewer is not fully aware of the very significant complications of an individual metrological validation of the open-path channels of HAI, otherwise he would acknowledge that such an effort would be worth a full paper by its own. Luckily, the HAI concept allows – by design - during cloud free conditions an inflight cross-validation of the open path channels by referencing them to the extractive channels.

c) In the focus of a paper is a "metrological, pressure dependent, absolute validation"; we do not agree that one has to explain all other developments which one has done over the last years. HAI is for that publication not relevant and therefore not mentioned. (Usually, reviewers criticize authors about "over-citing" themselves). The comment that both of the instruments measure with the same technique (I guess that is related to TDLAS spectroscopy) is true but not in any way a down site, as we have explained in the previous point. There are, beside the many referred academia devices, also commercial sensors such as the ones from Picarro, SpectralSensors, Siemens, ABB etc. which use this basic technology.

d) The reviewer claims that there was a "SEALDH-I"; this is actually true, but we never published an instrumental paper about this development step between SEALDH-0 (a non-flight qualified laboratory test setup) and a SEALDH-II (a fully featured airborne transfer standard). The only "publication" ever explaining SEALDH-I was short German handout [1] at a German meeting the "GMA/ITG-Fachtagung Sensoren und Messsysteme 2012" for industrial partners, which via the language barrier certainly doesn't penetrate the international community at all. Due to the concise structure of this handout, we highly doubt that anyone would gain any significant knowledge out of it, so we don't see the need to reference such almost grey literature (and again if we would, most reviewers would complain about referencing of non-english grey literature); the argument stays the same for the few press release about the SEALDH developments and flights.

Since we do not have any SEALDH-I instrumental paper, we cannot refer to a publication.

**To summarize:**
The reviewer commented: ". Instead the authors should include an extensive discussion of the relation between these instruments in the manuscript, not just the replies to the authors."
**This is NOT the focus of that paper at all.** The title and the abstract clearly state:
"SEALDH-II – a calibration-free transfer standard for airborne water vapor measurements:
**Pressure dependent absolute validation from 5 – 1200 ppmv at a metrological humidity generator**"

As a compromise to this reviewer, we can offer to modify the paper title to read as follows: "Pressure dependent, absolute validation of SEALDH-II – a calibration-free transfer standard for airborne water vapor measurements - from 5 – 1200 ppmv using a metrological humidity generator " or even ""Pressure dependent, absolute validation of a calibration-free, airborne laser hygrometer transfer standard (SEALDH-II) from 5 – 1200 ppmv using a metrological humidity generator"

*The second major issue I have with the comments is their relation to stratospheric measurements. The title includes "airborne water vapor measurements" and "5 – 1200 ppmv". Throughout the paper, the authors refer to a measurement range of 65 hPa – 950 hPa and that "the instrument is built for a concentration range from 3 - 40000 ppmv". The lower end of this pressure and mixing ratio range clearly includes stratospheric conditions. The authors motivate their work through significant differences between measurements, most of which are UTLS and stratospheric measurements. They discuss the AquiVIT campaigns, which were conducted largely for UTLS instruments to understand the difficulties of measuring stratospheric water vapor.*

This question is pretty similar to the question which the reviewer asked the last time.
I will try to answer these questions first in the briefest possible way; this repeats the paper and the rebuttal:
a) "the instrument is built for a concentration range from 3 - 40000 ppmv"
=> Yes: The lower range limit defined by offset uncertainty, the higher by condensation risks when dew point equals instrument temperature.
See line 164 – 169 in the most recent version of the script (first revised version)

b) "The lower end of this pressure and mixing ratio range clearly includes stratospheric conditions"
Yes, with the uncertainty stated in the paper (e.g. line 164 and 171); so at 3 ppmv the instrument has an uncertainty of 100% (!). Hence, we would not argue that SEALDH-II is a specialized UTLS instrument. If it had been the purpose, SEALDH-II's optical path length would have been certainly significantly longer.

c) "They discuss the AquiVIT campaigns, which were conducted largely for UTLS instruments to understand the difficulties of measuring stratospheric water vapor."
As written in the rebuttal extensively (and in the paper see from line 80), we referred to AquaVIT for two reasons: a) it is the one and only **representative airborne hygrometer comparison** ever done; if the reviewer knows about another one which is as representative as AquaVIT, **we would love to learn about it and reference it adequately**. b) AquaVIT covered a concentration range up to 150 ppmv. Especially the higher levels of the AQUAVIT overlap with the useful (=non offset-uncertainty driven) concentration range of SEALDH-II and allow an effective relation to SEALDH-II's validation results. Nevertheless, this does not mean that SEALDH-II is a specialized UTLS instrument.

[Figure]

For a more detailed discussion, we refer the reviewer back to our first rebuttal where we answered in greater detail the same question.

**One of the drivers for the AquaVIT campaigns was the disagreement between some aircraft and balloon borne observations. Water vapor observations of less than 5 ppmv were the most important range of this disagreement. Given the uncertainty of SEALDHII, this instrument would not contribute to this concentration range. Despite the un-questioned quality of the observations presented here, this raises the question of the importance of their results, especially given the frequent reference to AquaVIT. The authors should clearly point out, whether they have achieved a metrological validation of a field deployed instrument that can measure stratospheric water vapor (100 hPa, 5 ppmv) with an uncertainty of less than 5-10%.**
==> Thank you for that comment; nevertheless we are also a bit puzzled by it.
SEALDH is NOT an instrument designed especially for UT/LS application, nor is it one of the common "standard" even commercially available laser hygrometers like the ones from WVSS, Picarro, LosGatos, etc. Instead, SEALDH is designed as a metrological transfer standard which is still suitable for field applications and ensures full traceability to the primary water scale. Thus, the application range is wider than just UTLS and the requirements are also much more rigid and demanding. It is clearly stated in the abstract that we claim a "primary validation" that covers the relevant mid to upper tropospheric and lower stratospheric (MT/UT/LS) concentration and pressure range. This was done via "15 different $H_2O$ concentration levels between 5 and 1200 ppmv". At each concentration level, we studied the pressure dependence at 6 different gas pressures between 65 and 950 hPa." The introduction states (as well as the referred paper in section 2 [4]) that the instrument is built for a concentration range from 3 - 40000 ppmv; 0 -1000 hPa with a calculated uncertainty of 4.3% ± 3 ppmv. The lower detection limit is defined at that point where measurement value equals uncertainty (as explained in the paper: "The calculated mixture fraction offset uncertainty of ±3 ppmv defines the lower detection limit."). The measurements in figure 8 show how conservative these estimations for the calculated uncertainties were since the "real" deviations are only in the 0 - 20% range (with the lowest level at 35%). SEALDHs science and metrology targets are thus much wider and of different perspective than those of the AQUAVIT 1 or 2.
The reasons why we explained the AquaVIT study in such a detail are the following: Firstly, it is the most extensive comparison exercise which was carried out on such a level (representative for the community, externally reviewed, blind submission, clear statements, exemplary work, etc.) that allows a reliable relative performance statement about the state of the art of airborne hygrometry. Secondly, as you can see in the graph above, AquaVIT covered the range up to 150 ppmv. Most instruments deployed during AquaVIT had a smaller concentration range compared to SEALDH-II because they were specifically developed for stratospheric measurements. With reference to the reviewer's question, we have to comment that one should not compare apples with oranges. SEALDH-II is intended as a transfer standard for the troposphere AND lower stratosphere and thus designed as a wide range hygrometer. This asks for pragmatic compromises which have consequences. If we had in mind to make a comparison only for the stratospheric range, we could easily replace the 1 m long extractive cell with a fiber-coupled multipath cell having 10x - 50x more path length. This would drastically improve SEALDH-II's sensitivity AND offset stability in the low concentration range. As the sensitivity and offset largely scales with the cell path length, this would lead for SEALDH-II to a calculated uncertainty (see [4]) of 4.3% ± 0.3ppmv (@ 10 m) and of 4.3% ± 0.06ppmv (@50 m) respectively, which seems quite well suited for a stratospheric application.
It also has to be kept in mind that we stated conservative calculated uncertainties; the "real" experi-mental deviations are usually smaller, as described and shown in this paper. In our experience – also as participant in AV 1 and 2 - we made the experience that metrological humidity traceability options in the atmospheric sciences are more or less ignored, despite the availability of a fully validated metrological reference scale and infrastructure. It seems highly likely for us that an improved traceability would lead to better absolute accuracy of airborne hygrometry. SEALDH was designed to enable this link between metrology and field sciences. With the series of SEALDH papers and in particular with the one here under review, we intend to present the data that demonstrate this capability in a metrological sense and not just present another UTLS hygrometer.

So the argument of the reviewer that only improvements in stratospheric sensing would be required or relevant is a bit short sighted and ignores to a large extent our focus towards implementing general traceability for airborne hygrometry over the full range from LT to LS. Atmospheric science studies the entire atmospheric water cycle and not just its subsections in the stratosphere and as such, it would definitively be beneficiary to establish traceability with a wide range instrument such as SEALDH in order to improve the comparability between all atmospheric sub-compartments and not just between UTLS focused instruments.

We added a few lines to explain why we compare to AquaVIT.

The reviewer did not comment on this section in the first rebuttal or indicate why this discussion wasn't satisfactory. Maybe he overlooked it?

*All of this gives the impression that their instrument covers UTLS and stratospheric measurements, which in fact they do not. The change in wording in the revised manuscript is insufficient and the authors should add a discussion about what their results mean for the low end of the mixing ratio and pressure range for that instrument. After all they still claim their instrument to be an instrument for airborne measurements.*

We hope that we finally clarified that question with the text above.

Our current manuscript does -on purpose - NOT contain any specialized, dedicated UTLS / stratospheric claims; we described (e.g. 164) the range of SEALDH-II with (3 – 40000 ppmv). We truly hope that someone who knows the expression UTLS, also knows that 40000 ppmv or even the 1200 ppmv upper range covered in the present manuscript does NOT resemble a stratospheric water vapor mixing ratio. Thus again, we do NOT claim UTLS specialized performance or suitability for SEALDH-II

Maybe the reviewer got distracted by the sentence (line 55-59): "The ambient gas pressure (70 – 1000 hPa) and gas temperature (-80 – 40°C) ranges are large and both values change rapidly, the required $H_2O$ measurement range is set by the ambient atmosphere (3 – 40000 ppmv), mechanical stress and vibrations occur, and the sampled air contains additional substances from condensed water (ice, droplets), particles, or even aircraft fuel vapor (e.g. on ground)."

The typical humidity range of the atmosphere and the range of SEALDH-II is by accident the same (see above what drives SEALDH-II's limits).

*Given the experience of the authors I fully believe that they could build an instrument optimized for UTLS work as they explained in their replies.*

An instrument for UTLS would focus on the range between 0.1 – 50 ppmv (uncertainty 5% +/- 0.1 ppmv)). The setup for such an instrument would be pretty much different. Beside different laser wavelength/path length etc., the major challenges for such low water vapor mixing ratios are the sampling issues as well as the more rigorous parasitic water treatment. If the reviewer would like to cooperate in such a development, we would be more than happy to do so. SEALDH-II will never be able to achieve that low ppbv range, since it is as a wide range instrument 3 – 40000 ppmv (uncertainty 4.3% +/- 3ppmv), designed for a broad application range which certainly concentrates on the troposphere and not the stratosphere.

*However, the manuscript is about the instrument they have, not about an instrument they could potentially build. Since the current instrument is not suitable for UTLS work, it would be much more appropriate if the authors motivated their work with the appropriate instruments, i.e. exclusively tropospheric instruments and better discuss, which part of the atmospheric water vapor range the current instrument can measure and which it cannot.*

Yes, we cannot agree more! SEALDH-II's range is (3 – 40000 ppmv) it **not a UTLS** instrument. SEALDH-II can be used in all area where the measurements uncertainty is suitable (4.3% +/- 3ppmv). Therefore, if an uncertainty of e.g. 11% is acceptable, the instrument can be deployed at measurements of 30 ppmv.)

*In the discussion it would be appropriate to then discuss that the features of this instrument could be implemented into a new instrument dedicated to upper tropospheric and stratospheric water vapor measurements.*
The title of the publication is:
"SEALDH-II – a calibration-free transfer standard for airborne water vapor measurements:
Pressure dependent absolute validation from 5 – 1200 ppmv at a metrological humidity generator"
We don't believe the clarity of the paper increases if we extend it to a white paper of general instrument development discussions. That is not the focus of the paper at all and would risk that other reviewers than complain on a too broad and unspecific "focus".

*The authors have now also included a reference to SEALDH-I and its validation with the German national standard. Therefore, the validation exercise of SEALDH-II is not the first effort of that type.*
As mentioned above, there is no instrument publication of SEALDH-I. The comparison of SEALDH-0 (laboratory setup) with the primary standard [2], which **is** referred in the paper, was a) **NOT pressure dependent**, b) and at quite **high concentration levels** where the spectroscopic challenges are entirely different (keyword: saturation, high absorbance effects, linearity issues, condensation problems). The scope of this work is entirely different. There are clear explanations, why there has not been a "pressure dependent metrological validation", since the SI humidity is usually transferred at 1013 hPa. And there is also a clear reason why there was no metrological low humidity validation so far: Metrological low-humidity generators are usually operated with pure Nitrogen. This however changes the spectroscopy (N2 vs air broadening), so that a direct comparison with a primary standard is not possible.
So again – please – if the reviewer knows about any other "Pressure dependent absolute validation from 5 – 1200 ppmv at a metrological humidity generator", we would be very happy to read and include that.

**From our knowledge, this is the first "Pressure dependent absolute validation from 5 – 1200 ppmv at a metrological humidity generator".** This describes the novelty of the paper. Further, as this range coincides with as significant fraction of the atmospheric humidity range, we find that our paper is of broad interest for readers of AMT and thus well suited for a publication in AMT.

*This criticism is not against the value of their work, which I certainly recognize and appreciate. My criticism is that the authors do not put their work into the proper context. AquaVIT and the discussion of the accuracy of stratospheric and upper tropospheric measurements is not the proper context.*

We fully appreciate the discussion, but would also be very delighted if our arguments are recognized and discussed. For those readers which might also struggle to put our paper in the "right "context (which e.g. does neither include the second reviewer nor any of the numerous readers of the discussion paper, as there are no online comments), it is certainly very helpful that all reviewer discussions are also published with the manuscript. So the information to "clarify" this is certainly "nearby" and easily available. The idea to "treat" SEALDH-II as a specialized UTLS instrument was never our intention. This idea was entirely brought up by one of the two reviewers, we strongly objected to this idea from the first moment.

*Individual, detailed comments:*

*Please note that the line numbers refer to the final manuscript version without changes markup.*

*Lines 71ff: Statements such as "... those often show results which are not sufficient for validation of atmospheric models in terms of the required absolute accuracy, precision, temporal resolution, long-term stability, comparability, etc." are in that generality overstated. However, they were certainly true in the past for some sets of stratospheric measurements.*
=> We revised that

*Lines 80-81: Another major issue limiting the comparison of hygrometers in flight is the issue of how ambient air is moved to the instrument. Issues such as contamination and sampling line problems are additional issues, which have plagues in situ comparisons in the past. This needs to be discussed as this is also a limiting factor for SEALDH-II in atmospheric measurements. The authors nicely point out that contamination and line issues are handled carefully in laboratory calibrations by allowing instruments and air samples to equilibrate over long times. This is not possible in atmospheric measurements, the consequences of which need to be discussed.*

While we agree that is an issue for all (!) airborne measurements, this is a general operational problem when deploying any instrument into an environment where the measurement signal is temporarily fluctuating. However, we again have to stress that this paper focuses on "Pressure dependent **absolute validation** from 5 – 1200 ppmv at a metrological humidity generator" and not on any in-flight problems or field topics. Such problems are typically discussed in papers on filed campaigns. Therefore, we clearly focus here on an absolute validation (see line 81 and below). As a side note: The published peer reviewed information about the AquaVIT comparison also exclusively discussed static (!) i.e. "non-dynamic" concentration levels; a separation of the different influences is vital to identify the real cause of the deviations.

A dynamic characterization is not in the focus of this publication. Furthermore, there is - to our knowledge - no metrologically validated possibility to generate and provide a humidity generator capable to realize a metrologically defined "step function" to validate the temporal behavior of a humidity sensor

Generally, this is NOT specifically a limiting factor of SEALDH-II rather than a limiting factor for every extractive instrument. What's more, the "limitations" highly depend on the gas sampling system itself (e.g. type of inlet, pipe length, pipe coding, pipe temperature (heated?), flow, humidity level, etc.). Therefore, an accurate statement would only make sense if a full system was analyzed, which in the vast majority of papers on airborne hygrometers is NOT described in detail again due to the lack of suitable dynamic "step" reference generators.

=> We added a few words for clarification

*Line 149: "...The most holistic approach ..." This wording should be changed. Better wording is "...most extensive approach..."*
=> Changed

*Lines 152 ff.: Delete lines 152 – 154 and better write: "SEALDH-II is described in detail in Buchholz et al., (2016)"*

=> Revised

*Line 174: Here the authors should explicitly state that the uncertainty of the lower end of the mixing ratio measurement range at 3ppm is 100% due to the offset component. Here the authors need to be clear that this instrument is not suitable for lower stratospheric measurements and only somewhat useful in the upper troposphere compared to other instruments, which have a better claimed (although not validated) uncertainty.*

We wrote:
*"SEALDH-II's calculated linear part of the measurement uncertainty is 4.3%, with an additional offset uncertainty of ±3 ppmv" (…) SEALDH-II's measurement range covers 3 – 40000 ppmv. The calculated mixture fraction offset uncertainty of ±3 ppmv defines the lower detection limit."*

=> a) So we did exactly that - or did we misunderstood the reviewers comment?
=> b) We never (even if stated in the review many times) made the argument that SEALDH-II is in any kind a stratospheric instrument. On the contrary, we clearly stated e.g. in line 271 *"over a range which is particularly interesting for instruments on airborne platforms operating from troposphere to lower stratosphere where SEALDH-II's uncertainty (4.3% ± 3 ppmv) is suitable"*.
The word stratosphere is in the following contexts in the paper: a) explanation of the range of the atmosphere b) classification of typical operation ranges.
We have never and will never state that SEALDH-II is a stratospheric instrument.

*Line 183: At least except for the two other instruments, the authors already built. This sentence should be deleted to avoid confusion.*
Maybe there is a confusion with the line numbers: If it belongs to *"SEALDH-II's data treatment works differently from nearly all other published TDLAS spectrometers."* then this statement includes already the fact that only "nearly all" instruments do it differently. HAI and SEALDH-II are due to the first review so often cited in this paper that they just cannot be missed anymore - even if a reader reads just 20% of the publication.

*Lines 280f.: "…over a range which is particularly interesting for instruments on airborne platforms operating from troposphere to lower stratosphere where SEALDH-II's uncertainty (4.3% ± 3 ppmv) is suitable". Here the authors imply explicitly that the uncertainty of SEALDH-II is suitable for lower stratospheric measurements, where in fact the numbers in that same sentence clearly say the opposite. An uncertainty of 3 ppmv is not suitable for lower stratospheric measurements.*
We are a bit confused how this sentence can possibly be misunderstood. A reader who is not in the mindset of "forcing" SEALDH-II to be a stratospheric instrument will read the sentence as follows: a) Instrument has a range from 3 – 40000 ppmv with an uncertainty of (4.3% ± 3 ppmv). b) If he wants to measure 3 ppmv with that instrument, then he would need to use the provide uncertainty information, which yields a calculated uncertainty of 100% uncertainty at 3 ppm. This is exactly what we wrote and what the reviewer nevertheless requests?? By the way, SEALDH-II's "measured" value could still be much less noisy and imply a higher precision. But the uncertainty refers to the absolute accuracy and thus is defined by our statements as we targeted an "absolute validation". c) If he wants to measure 10 ppmv the uncertainty will be around 30%, at 100 ppmv around 5% and so on.
The part of the sentence "where SEALDH-II's uncertainty (4.3% ± 3 ppmv) is suitable" stated exactly that.

Without starting another lengthy discussion, the reviewer might also think about the difference between a "calculated uncertainty" and the "measured uncertainty/error/deviation". The data in figure 8 show that the "measured deviation" at 5ppmv and 100 hPa is just around 5-10% (and not 60% like the calculated uncertainty provides). Therefore, most instruments (which do not have the possibility to calculate an uncertainty due to the physical detection principle and flowing evaluation) would need to state this value (maybe with a scaling factor) as an uncertainty. Therefore - **we fully agree that SEALDH-II is NOT a stratospheric instrument** - it can be used in lower stratospheric conditions, however. But we cannot guarantee the absolute accuracy under this conditions from a metrological point of view, but most likely the deviations would be in the same range like during the validation (and not 100% at 3 ppmv). An uncertainty is NOT a typical deviation – this might another reason for the confusion.

*Lines 311-320: As said above, the core instruments of AquaVIT were built for upper tropospheric and stratospheric measurements in mind. In this sentence the uncertainties at the comfortable range for SEALDH-II are again compared to the challenging regime explored during AquiVIT, where SEALDH-II has itself a very large uncertainty. This is not appropriate and these sentences should be deleted.*
Here again the final graph of AquaVIT:

[Figure]

The range which is covered here is up to 150 ppmv. We presented in our manuscript measurements which are well overlapping with this concentration range, namely at: 100 ppmv, 75 ppmv, 50 ppmv, 35 ppmv, 20 ppmv, 15 ppmv, 10 ppmv, 5 ppmv; and at each concentration we measured 6 different pressure. ; therefore 48 different validation steps. The ranges of the instrument used during AquaVIT are partly up to several 1000 ppmv. Every hygrometer has a certain measurement range, so does SEALDH-II.

Therefore, we cannot understand this comment why this comparison is not valid from the reviewer's point of view and why we should exclude the reference to Aquavit. In particular, it should be noted here that we actually participated in AQUAVIT with the APICT instrument (mainly developed by our TDLAS group at Heidelberg University) and that all the data from APICT were evaluated by our group. Furthermore, I was the co-initiator of AQUAVIT, together with colleagues from KIT and FZJ. Thus, we have been very closely involved in AQUAVIT.

*Lines 375-378: This has been pointed out before and should be deleted here.*
It is not uncommon to skim over publications and this very important information should therefore be repeated here because it is vital for the assessment.

*Lines 389ff.: In the evaluation it is not acceptable to ignore the 65 and 125 hPa levels. The title of the paper clearly refers to airborne measurements; therefore, the validation effort needs to consider the atmospheric distribution of water vapor and not discuss how the validation result might look like without these levels.*

*"If one ignores the 65 hPa and 125 hPa measurements, which are clearly affected by higher order line shape effects (see above), the other measurements fit fairly well in a ±1 ppmv envelope function (grey)."*

   a) The sentence started with "if", which means that the second part of the sentence is only true if the first part is true as well.
   b) We describe that this effect is visible; we do not infer any(!) performance data which are related to any "full performance statements". Higher order lines shape effects (which are here nicely visible in figure 8) are in spectroscopy a commonly seen artefact if the Voigt line shape profile is used for spectral analysis.

*Line 403: Delete "To prevent further interpretations"*
*=> done*

*Lines 405 ff: better "This is an essential difference between calibration free instruments and …"*
*=> revised*

*Line 407: better "SEALDH-II tries to guarantee …"*
*The authors can never guarantee that ALL effects have been considered. They can only guarantee that all effects they know about have been considered.*
*=> revised*

*Line 428 f: Delete "novel" and "holistic"*
*=> I hope we clarified that in the text above that a) SEALDH-II's concept is novel (even if the publication is not about the instrument's setup) and b) the control of the "core spectrometer" is intense - compared to standard TDLAS spectrometer. Therefore, I guess the comment is obsolete.*

*Lines 451 ff: I would hope that the authors will as well validate HAI, which has been built by them and which is much more suitable for atmospheric measurements. Given the use of that instrument, validating it would be more than useful.*
=> Our main goal is to valid a concept which is deployed in different instruments. The instrument by itself is then just a representation of the concept. Therefore, several validation results are directly transferable to HAI, in particularly to the HAI's extractive signal channels.

*Figure 6-8: I maintain my earlier comment that the abscissa should be log(P), since the authors write about an instrument for airborne use, not laboratory use. For many atmospheric measurements, in particular aircraft based measurements, it is much easier to relate to altitude which varies nearly linear with log(P) than with pressure itself. This emphasizes the entire tropospheric distribution better than linear pressure.*

We would agree if SEALDH-II was presented here as a stratospheric instrument, which is the omnipresent idea of the reviewer. We think that the proposed graphical representation would really make the "main results graphs" much more difficult to understand. We will put the graphs with a log (p) scale here; so everyone can find the graphs in this representation in the discussion part.

[Figure]

[Figure]

[Figure]

[1]     B. Buchholz, N. Böse, S. Wagner, and V. Ebert, "Entwicklung eines rückführbaren, selbstkalibrierenden, absoluten TDLAS-Hygrometers in kompakter 19" Bauweise," in *AMA-Science, 16. GMA/ITG-Fachtagung Sensoren und Messsysteme 2012*, (2012), pp. 315–323, doi:10.5162/sensoren2012/3.2.3.

[2]     B. Buchholz, N. Böse, and V. Ebert, "Absolute validation of a diode laser hygrometer via intercomparison with the German national primary water vapor standard," *Applied Physics B*, vol. 116, no. 4, pp. 883–899, (2014), doi:10.1007/s00340-014-5775-4.

**Bernhard Buchholz[1,2,4], Volker Ebert[1,2,3]**

[revised manuscript text omitted]

Webster et al., 2004; Zöger et al., 1999a, 1999b) (non-exhaustive list). While for some atmospheric questions the quality level of the data often is sufficient (e.g. typically climatologies), there are also a variety of questions, especially validation of atmospheric models, where the required absolute accuracy, precision, temporal resolution, long-term stability, comparability, etc. needs to be higher. These problems can be grouped into two major categories: accuracy linked problems and time response linked problems. The latter is particularly important for investigations in heterogeneous regions in the lower troposphere as well as for investigations in clouds. In these regions, even two on average agreeing instruments with different response times yield local, large, relative deviations on the order of up to 30% (Smit et al., 2014). Important to keep in mind is that the total time response of a system is a superposition of the instrument's time response plus the sampling system's time response which typically depends on many parameters (e.g. type of inlet, pipe length, pipe coding, pipe temperature (heated?), flow, 
[revised manuscript text omitted]

---

## Author Response (AR3)

We thank the editor for carefully reading our manuscript and for the feedback aimed at helping us to further improve the manuscript.

**Comments to the Author:**

*Dear authors,*

*Please integrate your arguments from the rebuttal into the manuscript to avoid misinterpretations:*

*- connection to Aquavit campaign*
=> We added some sentences for clarification

*- reference to other instruments from same group if applicable*
=> We reviewed the references and the description of the instrument family

*- (possible) application of SEALDH-II for UTLS measurements*
=> We added information in conclusion and outlook

*- use of the terms "new" and "holistic"*
=> We revised and explained these terms in the paper.

*Potential unclear issues spotted by the editor:*

*- mixing ratio vs. mole fraction*
=> We clarified that in the paper

*- use of "mid" and "upper" atmosphere*
=> We clarified that in the paper at a metrological humidity generator ¶

[revised manuscript text omitted]

---

## Author Response (AR4)

We thank the editor for carefully reading our manuscript and for the feedback aimed at helping us to further improve the manuscript. We added all suggestions.